

# age_flow_line-1.0: a fast and accurate numerical age model for a pseudo-steady flow tube of an ice sheet

Frédéric Parrenin[1], Ailsa Chung[1], Carlos Martín[2]

[1]Univ. Grenoble Alpes, IRD, CNRS, INRAE, Grenoble INP, IGE, 38000 Grenoble, France
[2]British Antarctic Survey, Cambridge, UK

*Correspondence to*: Frédéric Parrenin (frederic.parrenin@univ-grenoble-alpes.fr)

**Abstract.** Numerical age models are useful tools for investigating the age of the ice in an ice sheet. They can be used to date ice cores or to interpret isochronal horizons which are observed by radar instruments. Here, we present a new numerical age
model for a flow line of an ice sheet. The assumption here is that the geometry of the flow line and the velocity shape functions are steady (i.e. constant in time). A time-varying factor can only be applied to the surface accumulation rates and basal melting rates. Our model uses an innovative coordinate system $(\pi,\theta)$, previously published, which is suitable for solving transport equations. Using this coordinate system, solving the age equation is simple, fast and accurate, because the trajectories of ice particles pass exactly through the nodes of the grid. Our numerical scheme, called Eulerian-Lagrangian, therefore combines
the advantages of Eulerian and Lagrangian schemes. We present an application of this model to the flow line going from Dome C to Little Dome C and show that horizontal flow is a non-negligible factor which should be considered when modelling the age-depth relationship of the Beyond EPICA ice core. The code we developed for age modelling along a flow tube is named age_flow_line-1.0 and is freely available under an open-source license.

## 1 Introduction

Ice sheets, such as the current Greenland or Antarctic ice sheets, or the past Fennoscandian or Laurentide ice sheets, are a fundamental component of the global climate system (Oerlemans and Van Der Veen, 1984). They are the most important reservoir of fresh water on Earth, their existence causes global sea level to be lower than it would otherwise be, their white surfaces lead to a strong albedo which cools the regional and even global climates, their high elevation modifies the atmospheric circulation and the fresh water and icebergs they eject into the oceans can perturb the deep ocean circulation.
Therefore, it is essential to understand the processes and the boundary conditions which govern the evolution of ice sheets.

Current ice sheets are also an extraordinary archive of past climates, as they can record local temperature variations when the snow fell (Jouzel et al., 2007; NorthGRIP project members, 2004), the past atmospheric composition (Loulergue et al., 2008; Lüthi et al., 2008) and atmospheric impurities (Lambert et al., 2012). For the interpretation of the ice core records, it is essential to date the ice and reconstruct the trajectories and mechanical histories of ice particles within the ice sheets.



Modeling the age of the ice in ice sheets is important for several reasons. This includes dating existing ice cores (Buchardt,
2009; Huybrechts et al., 2007; Johnsen and Dansgaard, 1992; Parrenin et al., 2007) and investigating new potential ice core
drill sites (Chung et al., 2023; Obase et al., 2023; Parrenin et al., 2017; Van Liefferinge and Pattyn, 2013). Age observations
from ice cores (Bouchet et al., 2023; Oyabu et al., 2022) or from radio echo sounding (Cavitte et al., 2016; MacGregor et al.,
2015) can help constrain the boundary conditions of ice sheets or understand the internal processes of ice sheet models using
inverse methods.

Various type of numerical models with a transport scheme to compute the age within the ice sheets have been implemented.
For example, large scale transient models have been used (Lhomme et al., 2005a; Sutter et al., 2019). The advantages of these
models are that they are fully transient and consider many of the different physical processes involved in ice sheet evolution.
However, boundary conditions such as the geothermal flux, the basal drag or the surface accumulation rates for these models
are poorly known. Due to their large computation time, the evolution equations are often simplified and their resolution is
multi-kilometric, meaning that detailed bedrock relief is not/ cannot be taken into account. Moreover, these models might
simulate a present-day surface topography which is different from the observed surface topography, making it challenging to
compare the model with existing ice core or radar observations. To improve the resolution or the representation of physical
processes in these models, it is possible to embed a local high resolution and high order model within a simplified continental
ice sheet model (Huybrechts et al., 2007).

Another approach is to use local 1D (purely vertical) or 2.5D (flow tube) models with a prescribed geometry. These models
can be either transient (Koutnik et al., 2016; Obase et al., 2023), steady state (Waddington et al., 2007) or pseudo-steady
(Chung et al., 2023; Parrenin et al., 2017) which is a steady model but with a scaling of the age to account for the temporal
change of surface accumulation rate.

There exist several numerical scheme for solving the transport equations, in particular the age equation (Rybak and Huybrechts,
2003). The Lagrangian scheme follows ice particles within their journey and is generally stable. However, the tracers are
dispersed and it is not possible to know a priori where these tracers will end up at the end of the simulation. The Eulerian
scheme solves the age equation on a grid and does not have the problem of tracer dispersion but can be numerically unstable,
especially in 3D (Hindmarsh et al., 2009). The third numerical scheme is the semi-Lagrangian scheme, which, as with the
Eulerian scheme, solves the age equation on a grid but is more stable than the Eulerian scheme (Lhomme et al., 2005b; Sutter
et al., 2021).

Here we present a new 2.5D (flow tube) pseudo-steady numerical age model, named age_flow_line-1.0. This numerical age
model uses an innovative coordinate system $(\pi,\theta)$ previously published (Parrenin and Hindmarsh, 2007) and which is suitable
for solving transport equations. We solve the age and transport equations using a Eulerian-Lagrangian scheme that uses an
analytical derivation of trajectories and a grid which tracks these trajectories. In Section 2, we first present the analytical and





numerical foundations of the model and its implementation. In Section 3, we show an application of the model to the flow line between Dome C and Little Dome C in Antarctica. In Section 4, we discuss the advantages and limitations of the model.

## 2 Description of the model

### 2.1 Notations and analytical aspects

We first consider a steady flow line of an ice sheet. This means that we assume that factors such as the geometry of a flow line (eg. ice thickness) and the vertical shape function do not change in time. In this model, we will consider non-steady accumulation rate and melting rate, but this will be discussed later. The notation and equations follow that of Parrenin and Hindmarsh (2007) but the assumptions are slightly relaxed since we allow the flow tube width and relative density to vary vertically. We write the equations in the $(x,z)$ coordinates where $x$, the horizontal coordinate along the direction of ice flow,

is the distance from the dome and $z$ is the vertical coordinate. We suppose that the horizontal direction of the flow does not depend on the vertical position and is time-independent, and we represent the lateral flow divergence by the varying flow tube width $Y(x,z)$. In practice, for grounded ice and under the shallow ice approximation, the direction of the flow can be determined from the surface elevation gradient. In this case, the steady assumption for the flow tube means that the shape of surface elevation contours does not change with time. The direction of the flow can also be determined by surface velocity

measurements, as is done for the modelling of the flow line from Dome C to Little Dome C (Chung et al., 2024). Using surface velocity measurements is more appropriate where the surface is flat because in this case, the flow might not follow the greatest slope.

The ice-sheet geometry is given by the bedrock elevation $B(x)$, the surface elevation $S(x)$ and the total ice thickness

$H(x) = S(x) - B(x)$. We also denote the depth in the ice sheet by $d$. We let snow/ice relative density be $D(x,z)$ and the

surface accumulation of ice and basal melting rate at the ice–bedrock interface be $a(x)$ and $m(x)$, respectively. We let $u_x(x,z)$ denote the horizontal velocity and $u_z(x,z)$ the vertical velocity of the ice particles. We also let $\chi(x,z)$ represent the age of the ice particle.

We now define fluxes used to derive the stream function. The partial horizontal flux $q_H(x,z)$ is defined as the horizontal flux passing below depth $z$:

$$q_H(x,z) = \int_B^z Y(x,z')D(x,z')u_x(x,z')dz',$$

(1)

with $Q_H(x) = q_H(x,S)$ being the total horizontal flux at position $x$. We further define the basal melting flux as:



$$Q_m(x) = \int_0^x Y^B(x')m(x')dx'. \tag{2}$$

where $Y^B(x) = Y(x,B)$ is the tube width at the bedrock.

Because of the steady-state assumption, the two components of the velocity field can be derived from one scalar variable called the stream function, denoted by $q(x,z)$ which follow these equations:

$$YDu_x = \frac{\partial q}{\partial z},$$

$$YDu_z = -\frac{\partial q}{\partial x}. \tag{3}$$

The flux through any path linking two points $A$ and $B$ is independent of the chosen path and is the difference $q_B - q_A$. Therefore, $q = Q_m + q_H$. By definition, the contours of the stream function correspond to the trajectories of ice particles.

We define the total flux at position $x$ by $Q(x) = q(x,S)$, except at the dome where $Q(x) = 0$. We define the normalized stream function $\Omega(x,z)$ such that the stream function $q$ is given by

$$q(x,z) = Q(x)\Omega(x,z) \tag{4}$$

This new variable can be related to the horizontal flux shape function $\omega$ through

$$\Omega(x,z) = \frac{\omega(x,z) + \mu}{1 + \mu} \tag{5}$$

with $\omega$ defined by (Parrenin et al., 2006; Reeh and Paterson, 1988):

$$q_H(x,z) = Q_H(x)\omega(x,z), \tag{6}$$

and with $\mu$ being the ratio of the melting flux to the horizontal flux

$$\mu(x) = \frac{Q_m(x)}{Q_H(x)}. \tag{7}$$

In order to determine particle trajectories and the age of ice, we can now transform the $(x,z)$ coordinate system to a new system $(\pi,\theta)$ defined as (Parrenin and Hindmarsh, 2007):

$$\pi(x) = \ln\left(\frac{Q(x)}{Q^{\text{ref}}}\right)$$

$$\theta(x,z) = \ln(\Omega(x,z)) \tag{8}$$



where $Q^{\text{ref}}$ is a reference flux (in practice, we use the total flux at the downstream boundary of the domain). Note that the change of variable from $x$ to $\pi$ requires $Q(x)$ to be an increasing function, and we must therefore assume that the accumulation $a$ is strictly positive all along the flow-line. Also, the change of variable from $z$ to $\theta$ requires $\Omega$ to be an increasing function of $z$, which corresponds to assuming that there is no reverse flow.

We then define the horizontal and vertical velocity components in this new coordinate system: $u_\pi = \frac{d\pi}{dt}$ and $u_\theta = \frac{d\theta}{dt}$. Parrenin (2013) showed that these velocity components can be written as

$$u_\pi = \frac{Y^S a}{YD} \frac{\partial \Omega}{\partial z},$$

$$u_\theta = -\frac{Y^S a}{YD} \frac{\partial \Omega}{\partial z}. \tag{9}$$

where $Y^S(x) = Y(x,S)$ is the tube width at surface. This way, it can immediately be seen (Parrenin and Hindmarsh, 2007), that these velocity components are opposite and therefore, trajectories in the $(\pi,\theta)$ coordinate system are straight lines of slope -1 (see Figure 2).

Integrating the time spent along the trajectory from Eq. (9), the age of an ice particle can be written as

$$\chi = \int_{\pi_0}^{\pi} \kappa \, d\pi \tag{10}$$

with $\pi_0$ the initial value of $\pi$ when the ice particle was at surface and with the $\kappa$ parameter defined as:

$$\kappa = \frac{YD}{Y^S a} \frac{\partial z_\Omega}{\partial \Omega}. \tag{11}$$

We can now define the vertical thinning function, which is the ratio of the vertical thickness of a layer to its initial ice equivalent vertical thickness when it was at surface. Using this $(\pi,\theta)$ coordinate system, it can also be shown that the vertical thinning function can be written (Parrenin, 2013):

$$\tau = \Omega \frac{Y^S a}{YD} \frac{D_0}{a_0} \left( 1 - \frac{1}{\kappa} \int_{\pi_0}^{\pi} \frac{\partial \kappa}{\partial \pi} \right)^{-1} \tag{12}$$

where $D_0$ and $a_0$ are respectively the relative density and accumulation rate when the snow/ice particle was deposited.

We can now consider variations of surface accumulation and basal melting rates and assume they have common temporal variations (Parrenin et al., 2006; Parrenin and Hindmarsh, 2007):





$$a(x,t) = \bar{a}(x)R(t) \tag{13}$$

$$m(x,t) = \overline{m}(x)R(t)$$

where $\bar{a}(x)$ is the average accumulation rate and $\overline{m}(x)$ is the average melting rate at position $x$ and $R(t)$ is a positive temporal

multiplicative factor. Using this assumption, the trajectories and isochrones are the same as those of the steady problem which

uses $\bar{a}$ and $\overline{m}$, but velocities of ice particles are enhanced or reduced depending on the value of $R(t)$. It is possible to deduce

the real age of ice particles $\chi$ from the steady age $\bar{\chi}$ using the following change of time variable

$$\bar{t} = \int_0^t R(t)dt, \tag{14}$$

where $\bar{t}$ is the steady time and $t$ is the real time.

**2.2 Simplifying assumptions**

To simplify, we assume that snow is compressed instantaneously into ice at surface. In practice, we convert real depth into ice

equivalent depth using the observed relative density profile. Moreover, we assume that the flow tube has vertical walls, that

is, $Y$ does not depend on $z$. This way, the equations for $\kappa$ and $\tau$ become:

$$\kappa = \frac{1}{a}\frac{\partial z_\Omega}{\partial \Omega}. \tag{15}$$

$$\tau = \Omega \frac{a}{a_0}\left(1 - \frac{1}{\kappa}\int_{\pi_0}^{\pi}\frac{\partial \kappa}{\partial \pi}\right)^{-1} \tag{16}$$

**2.3 Numerical aspects**

The $(\pi,\theta)$ coordinate system is very useful for solving any transport equation and in particular the age equation. Indeed, if we

define a regular grid in $(\pi,\theta)$ with the same step $\Delta\pi = \Delta\theta = \Delta$, the trajectories of ice particles pass exactly through the nodes

of the grid (Figure 1). Therefore, starting from the surface where the age is $\chi = 0$, it is possible to deduce the age on the

horizontal grid line below where $\theta = -\Delta$ by calculating the time needed for each ice particle to cross a cell. Then the age on

the horizontal grid line below ($\theta = -2\Delta$) can be calculated, etc. As particles move downstream, the conditions at the upstream

boundary should be known. As the $\pi$ scale is logarithmic, there is a singularity at the dome. Therefore, the horizontal $\pi$ scale

starts at a point slightly downstream of the dome, where an age can be prescribed, for example using a dome solution (which

assumes purely vertical movement).





The time needed to cross a cell is

$$\Delta\chi = \int_{\pi}^{\pi+\Delta} \kappa d\pi. \tag{17}$$


Here, we assume that $1/a$ is continuous and varies linearly between two consecutive $\pi$ nodes. Similarly, we assume that $z_\Omega$ is continuous and varies linearly between two consecutive $\theta$ nodes along a vertical grid line (Figure 1). It means that $\frac{\partial z_\Omega}{\partial\Omega}$ is a stepwise function along $\theta$ and a continuous and linear by parts function along $\pi$. Integrating $\kappa$ along $\pi$ across a cell therefore corresponds to integrating a second order polynomial.

Similarly, we can calculate the vertical thinning function using Eq. (11) by integrating the following quantity along each cell:

$$\int_{\pi}^{\pi+\Delta} \frac{\partial\kappa}{\partial\pi}d\pi. \tag{18}$$

With our assumption that $1/a$ and that $z_\Omega$ are piecewise linear functions, $\partial\kappa/\partial\pi$ is a first order polynomial between $\pi$ and $\pi + \Delta$, so it can be easily integrated.

To calculate the initial horizontal positions, the total flux or the initial accumulation rate when the ice particles were at surface,
we must transport these quantities from the surface along the diagonals in the $(\pi,\theta)$ grid (Figure 2). The trajectories of the particles are given by the iso-contours of the stream function $q$.

In our age_flow_line-1.0 code, it is possible to determine the age, accumulation and thinning function on a refined vertical grid corresponding to an existing or potential ice core. We first determine the $\pi$ coordinate of the ice core and calculate the age and thinning profiles by a weighted average of the two adjacent vertical profiles of the 2D grid. We then interpolate this coarse
1D vertical profile onto a refined 1D ice core profile. For the thinning function, we perform a linear interpolation while, for the age, we use a quadratic interpolation.

In order to verify that the uncertainty due to interpolation is negligible, we also use a different method to estimate the thinning function and age at the ice core location. Once the age, thinning and the initial accumulation rate are calculated for an ice core, it is possible to derive the age to get another estimate of the thinning function, or to integrate the layer thickness to get another
estimate of the age. We then check that the two estimates of age and thinning (one integrating the age and the other integrating the thinning) are consistent within numerical uncertainties.





### 2.4 Programming aspects

age_flow_line-1.0  is entirely written using the Python-3 programming language with several scientific modules (numpy, scipy, matplotlib). age_flow_line-1.0 both solves the transport equations and displays the results as figures. The figures displayed for the 2D grid are: the 2D grid in the $(x,z)$ and $(\pi,\theta)$ coordinate system, the age in the $(x,z)$, $(x,d)$ and $(\pi,\theta)$ coordinate systems, thinning in the $(x,z)$ coordinate system, obtained either by direct integration of the thinning or by differentiation of the age, the $\omega$ function in the $(x,z)$ coordinate system, the stream function $q$ and its contours (the stream lines, which represent particle trajectories) in the $(x,z)$ coordinate system. Along the $x$ coordinate of the flow line, figures display the accumulation rate, the basal melting rate, the width of the flow tube and the surface velocity or the total flux $Q$. Along the 1D vertical grid of each ice core, one figure displays the accumulation and layer thickness as a function of the age, and another figure displays the age, spatial origin and thinning as a function of depth.

For the horizontal flux shape function $\omega$, we use the Lliboutry profile (Lliboutry, 1979; Parrenin and Hindmarsh, 2007):

$$\omega(\zeta) = 1 - \frac{p+2}{p+1}(1-\zeta) + \frac{1}{p+1}(1-\zeta)^{p+2} \tag{19}$$

where $\zeta$ is the normalised vertical coordinate (0 at the bedrock and 1 at the surface). Therefore, only the value of $p$ needs to be defined for each horizontal position $x$ along the flow line. However, one could easily implement another type of horizontal flux shape function, for example the Dansgaard-Johnsen profile (Johnsen and Dansgaard, 1992).

For the upstream boundary condition on the age, we impose a dome profile:

$$\chi = \int_{\theta}^{0} \kappa d\theta. \tag{20}$$

The core of the code is entirely separate from the experiment directory where the results of the run are saved. The experimental directory is composed of a general parameter file in the YAML format (age_flow_line-1.0 uses the pyaml module) and text files that define the accumulation rate $a$, basal melting rate $m$, width of the flow tube $Y$, surface elevation $S$, ice sheet thickness $H$ and exponent of the Lliboutry profile $p$. There is also a vertical density profile given in a text file for transforming real depths to ice equivalent depths and this profile is assumed to be the same along the whole flow line.





## 3 Application to the DC-LDC flow line

To demonstrate the performance and ability of our age model, we apply it to the flow line between Dome C and Little Dome C (East Antarctica). The flow tube has already been determined in the companion paper Chung et al. (2024). It is ~40 km long with a strong lateral flow divergence because we are on a divide (Figure 3). The parameters used for our simulations, namely the surface accumulation rate, basal melting and Lliboutry parameter, are those from Chung et al. (2023) obtained by fitting a 1D pseudo-steady age model onto observed isochrones. This 1D model finds a best fit value for a mechanical ice thickness

and uses the comparison with the observed ice thickness to infer basal conditions. It is therefore the mechanical ice thickness that we use for our bottom boundary condition here. The aim of our simulation is therefore to estimate whether horizontal advection is an important mechanism to take into account in the modelling of the age along this flow line. This simulation is purely for demonstration purposes and should not be seen as realistic. This is because we are using the results of the 1D inverse model for a 2.5D flow tube model, instead of optimizing the 2.5D model directly onto the observed isochrones and ice core

datasets. The optimization of this 2.5D model is the scope of the companion paper Chung et al. (2024).

The boundary conditions of the model are plotted in Figure 4. The meshes of the model are plotted in Figure 5. The horizontal flux shape function ω is plotted in Figure 6. The modelled age, trajectories and vertical thinning function are plotted in Figure 7, Figure 8 and Figure 9. The results for the next ice core at BELDC are plotted in Figure 10. These figures (from 4 to 10) are automatically generated by the age_flow_line-1.0 software.

On Figure 5, it can be observed that the mesh is refined near the bedrock. This is expected, since the grid is exponential with respect to the flux shape function $\Omega$. When there is no basal melting (which is the case in our application with the mechanical ice thickness), the grid does not extend down to the bedrock but approach it asymptotically. The horizontal resolution also varies in our application, but several parameters affect this resolution. One would expect that the horizontal resolution would decrease when the distance to the dome increases, since the grid is exponential in total flux $Q$. But this is partially compensated

by the exponential increase of the tube width along the flow line.

From Figure 8, it can be observed that the ice particles may originate >20 km upstream along the divide. Therefore, horizontal flow is not negligible along this flow line and should be taken into account when modelling the age of the ice. In particular at BELDC (Figure 10), the bottom ice originates up to ~22 km upstream from our modelling.

## 4 Discussion

### 4.1 Numerical aspects

This age model is accurate and with minimal numerical diffusion since quantities are transported from one node to the next without interpolation. This could help test the accuracy of numerical schemes of more complex 3D and transient models.





This model has similar complexity of a steady model but considers the variations of accumulation with time which is non-negligible along a glacial-interglacial cycle (approximately a factor of 2 on the East Antarctic plateau). For a 1000x1000 grid, the computation time is ~0.1 s on a recent laptop computer for the numerical calculations, with a few additional seconds for the building of the figures. For this grid, the memory footprint of the simulation is ~0.8 Gb.

Due to its fast computation time, this model is appropriate for inverse simulations where the results are fitted to observations. Indeed, inverse simulations require multiple runs of the forward model to find the optimal set of parameters, which can be computationally expensive.

This model could also be appropriate for representing the advection term in a heat equation, although the diffusion term is better dealt with in a physical (x,z) coordinate system.

### 4.2 Modelling of the DC-LDC flow line

We have performed a simulation where both vertical and horizontal ice flow is taken into account. We show that it has a non negligible effect on the age modelling of the Beyond EPICA ice core, with particles originating sometimes >20 km from their current position. Our simulation is provided as a proof of concept for the model we have developed. However, it is not appropriate to use the result of the 1D inverse model to determine the parameters of the 2.5D model. Furthermore, we can see in Figure 7, that modelled isochrones (in black) do not agree well with the observed isochrones (in white). For a more realistic simulation, the parameters of the model should be optimized so that modelled age fits the radar and ice core age observations. This is the purpose of the companion paper of Chung et al. (2024).

### 4.3 Limitations of the pseudo-steady assumption

Our model relies on the pseudo-steady assumption, which states that all variables are in steady-state except for a temporal multiplicative factor which is both applied to accumulation and melting.

First, the geometry is assumed to be in steady-state. This assumption is adapted for the interior of East Antarctica, where relative variations in ice thickness were small (Ritz et al., 2001), but it still requires that the flow lines have not varied too much in the past, which is still unclear. Greenland and West Antarctica may have had a relatively stable geometry since the last glacial inception but they probably encountered important changes during the last interglacial.

Second, the same temporal multiplicative factor is applied to both surface accumulation and basal melting. There are no physical reason to assume surface accumulation and basal melting have varied in concert, this is just a mathematical convenience. But because basal melting is generally small in front of surface accumulation, this assumption should not be too dramatic if the right average-in-time basal melting is given as input to the model. Moreover, this temporal factor assumes that the surface accumulation spatial pattern has remained stable in time. This assumption relies on a stable snow precipitation process and a stable snow re-deposition by wind, which might not always be the case (Cavitte et al., 2018) .



Our pseudo-steady model can therefore be seen as a first approach to model the age field along flow lines at high resolution, but more complex 3D and transient models are necessary to fully investigate the age of the ice in ice sheets.

### 4.4 Possible improvements

In this implementation, we have made a few simplifying assumptions that could be relaxed in the future. First we have assumed that the flow tube has vertical walls, which is for example not the case along a ridge (Passalacqua et al., 2016). Second, we have converted real depths into ice equivalent depth using a unique density profile along the flow line, while clearly for long flow lines the density profile might change with the distance to the dome. Third, we have used a Lliboutry profile for the horizontal flux shape function $\omega$, while other profiles could be used. In particular at a steady dome where Raymond arches develop (Raymond, 1983), the Lliboutry profile does not seem to be suitable (Martín and Gudmundsson, 2012). Fourth, we prescribed a dome solution on the upstream boundary of the domain, but any boundary condition could be prescribed.

Due to the numerical simplicity of the model, it could be possible to derive its Jacobian, which would make inverse simulations more efficient.

### 5 Conclusions

We have developed a numerical model to calculate the age of the ice along a pseudo-steady flow tube of an ice sheet. Our Eulerian-Lagrangian scheme combines advantages of Eulerian and Lagragian schemes. There is a regular grid where the age is calculated, as in Eulerian schemes and there is no numerical diffusion, as in Lagrangian schemes. Our model is computationallyWith our assumption that 1/a and that z_Ω and linear piecewsie functions effective, which opens up new prospects for optimizing its parameters according to observations. We have applied our model to the DC-LDC flow line and we have shown that horizontal flow is non negligible there, with ice particles sometimes travelling >15 km which has implications for the age scale of the Beyond EPICA ice core. The next step is to optimize the parameters of the model according to age observations such as radar-observed isochrones and ice core dated horizons, which is done in the companion manuscript, Chung et al. (2024).

### Code availability

age_flow_line-1.0 is an open source model available under the MIT license. It is hosted on the github facility (Parrenin, 2024a) and the version corresponding to this submitted manuscript has been published on Zenodo (Parrenin, 2024b). The main author (F. Parrenin) can provide support for people who would like to use the software.



## Author contributions

FP developed the analytical formulas and the numerical code. AC tested the software and provided feedback. FP wrote a first version of the manuscript which was further commented and improved by AC and CM.

## Competing interests

The authors declare that they have no conflict of interest.

## Acknowledgements

This work was funded by the CNRS/INSU/LEFE projects IceChrono and CO2Role and by the French ANR project ToBE (ANR-22-CE01-0024). This publication was generated in the frame of the DEEPICE project. The project has received funding from the European Union's Horizon 2020 research and innovation programme under the Marie Sklodowska-Curie grant agreement no. 955750. This publication was generated in the frame of Beyond EPICA. The project has received funding from the European Union's Horizon 2020 research and innovation programme under grant agreement No. 730258 (Oldest Ice) and

No. 815384 (Oldest Ice Core). It is supported by national partners and funding agencies in Belgium, Denmark, France, Germany, Italy, Norway, Sweden, Switzerland, The Netherlands and the United Kingdom. Logistic support is mainly provided by ENEA and IPEV through the Concordia Station system. The opinions expressed and arguments employed herein do not necessarily reflect the official views of the European Union funding agency or other national funding bodies. This is Beyond EPICA publication number XX.

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






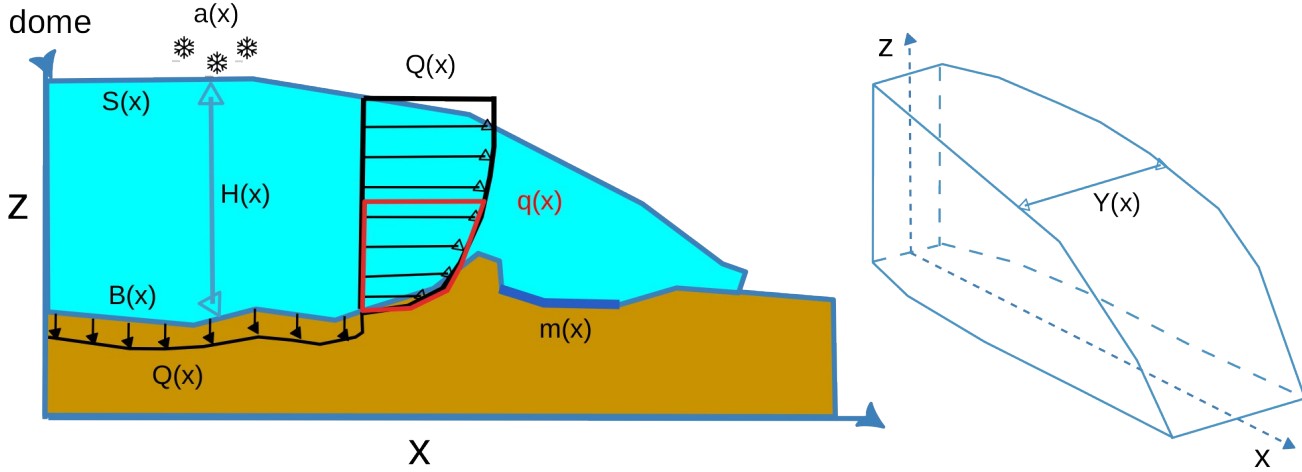

**Figure 1:** Diagram of the flow tube of an ice sheet with the principle notations used in this article.





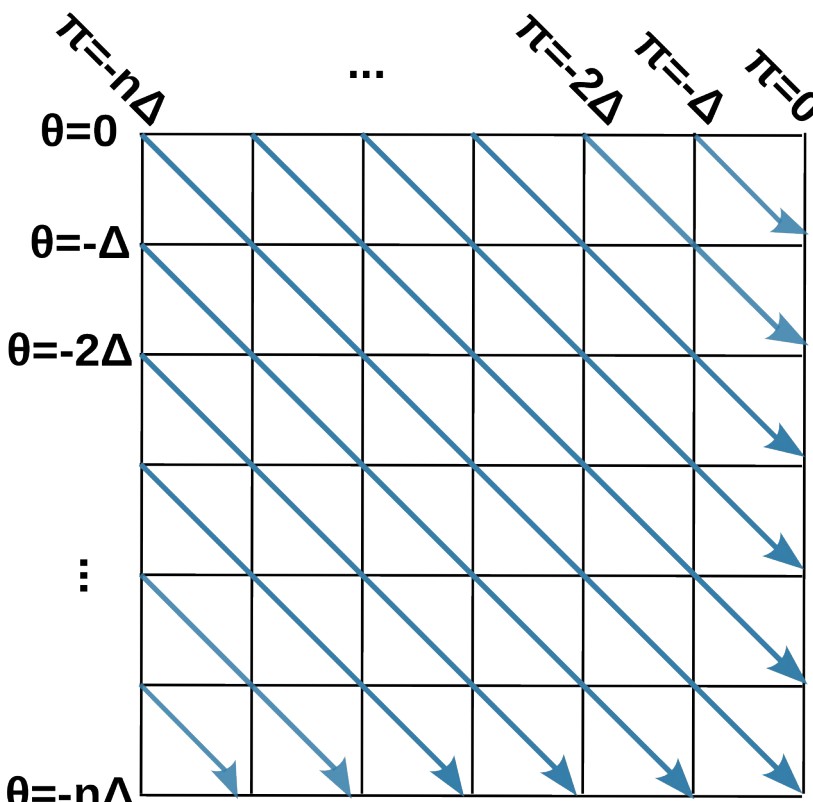

**Figure 2**: Diagram representing the $(\pi,\theta)$ grid (in black) and the trajectories (blue arrows) in this coordinate system.



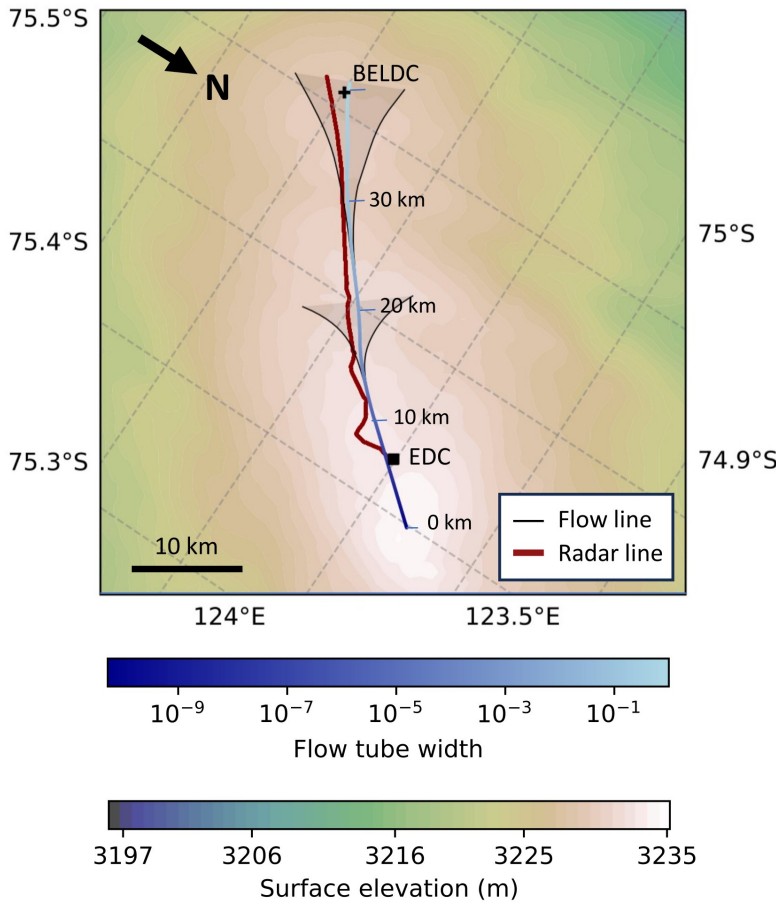

**Figure 3:** Map of the flow line (central blue line) going from DC to LDC with the width of the flow tube according to the blue color bar. Two pairs of adjacent flow lines ending at BELDC and 20 km downstream from the dome are also shown to emphasize the strong lateral flow divergence along this divide. The background colour represents the surface elevation. The dark red line represents the radar transect where the 1D inverse model of Chung et al. (2023) was applied. Adapted from Chung et al. (2024).





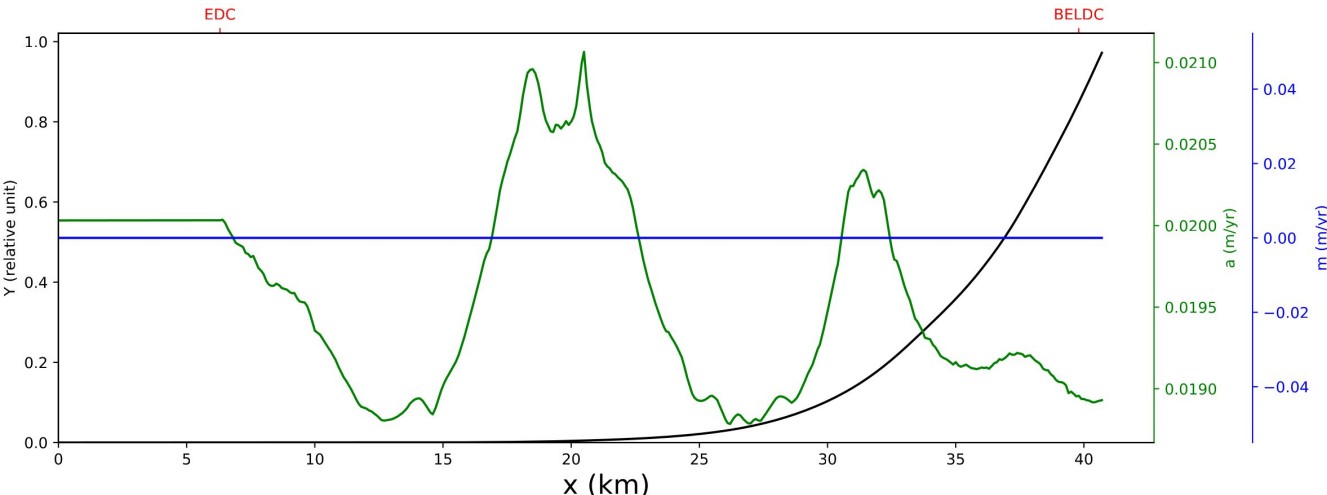

**Figure 4:** Boundary conditions of the model along the DC-LDC flow line: surface accumulation rate (green), basal melt rate (blue) and tube width (black). The position of the EDC and BELDC drilling sites are indicated in red on the top bar. This figure was automatically generated by the age_flow_line-1.0 software.







**Figure 5:** Mesh of the age_flow_line-1.0 model experiment in the $(\pi,\theta)$ (**top**) and (x,z) (**bottom**) coordinate system. The positions of the EDC and BELDC deep drill sites are plotted in dashed red. The observed bedrock is in thick black and the mechanical bedrock in violet. For better readability, the resolution of the grids has been decreased by a factor of 5. These figures were automatically generated by the age_flow_line-1.0 software.




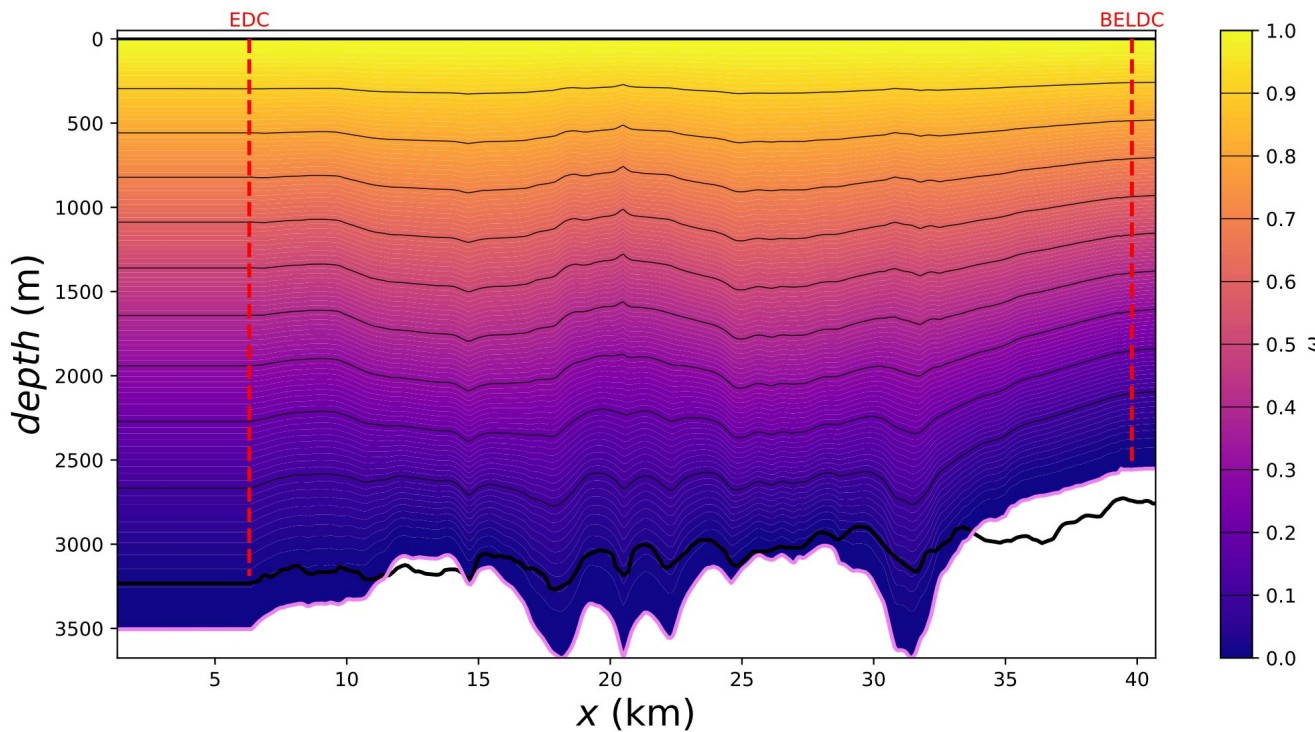

**Figure 6:** The $\omega$ horizontal flux shape function along the DC-LDC flow line. The positions of the EDC and BELDC deep drill sites are plotted in dashed red. The observed bedrock is in thick black and the mechanical bedrock in violet. This figure was automatically generated by the age_flow_line-1.0 software.




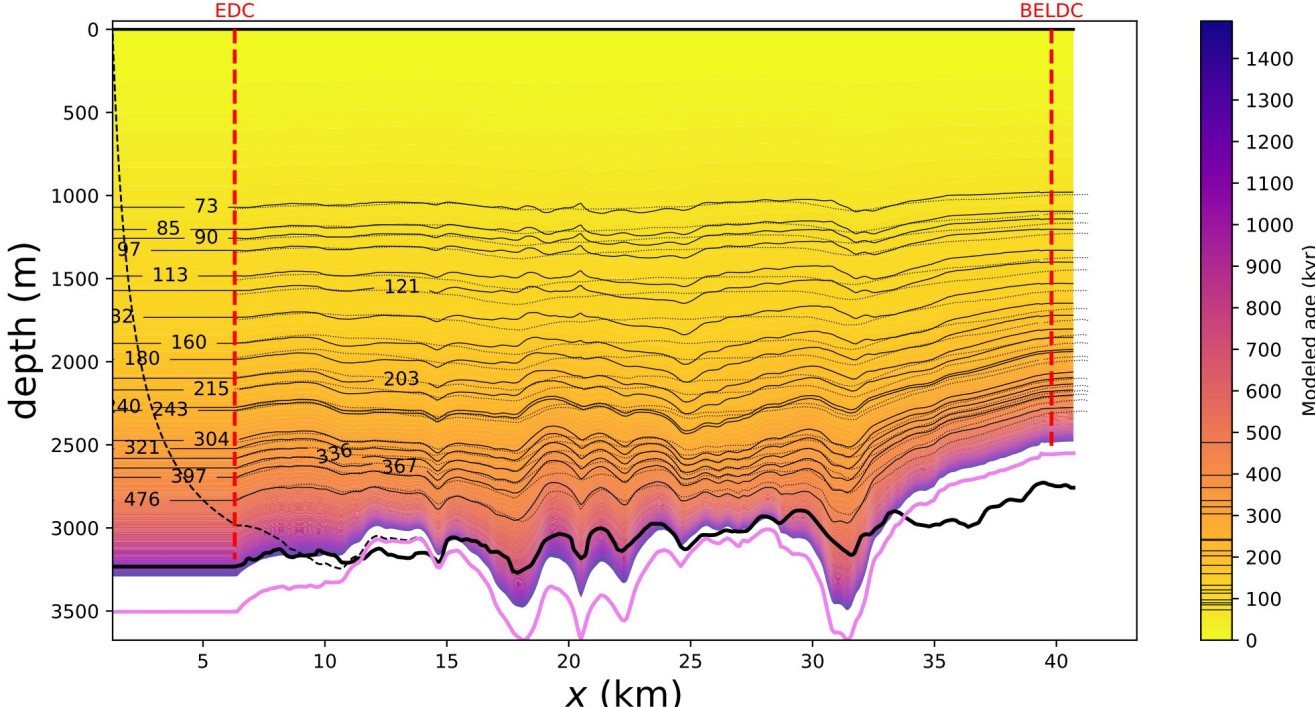

**Figure 7:** Modelled age along the DC-LDC flow line, according to the colour scale on the right. The modelled isochrones are plotted in solid black and their age is represented on the colour bar. The positions of the EDC and BELDC deep drill sites are plotted in dashed red. The black dashed line represents the trajectory originating from the surface upstream corner. The dotted black lines are the isochrones observed by radar. The observed bedrock is in thick black and the mechanical bedrock in violet. This figure was automatically generated by the age_flow_line-1.0 software.





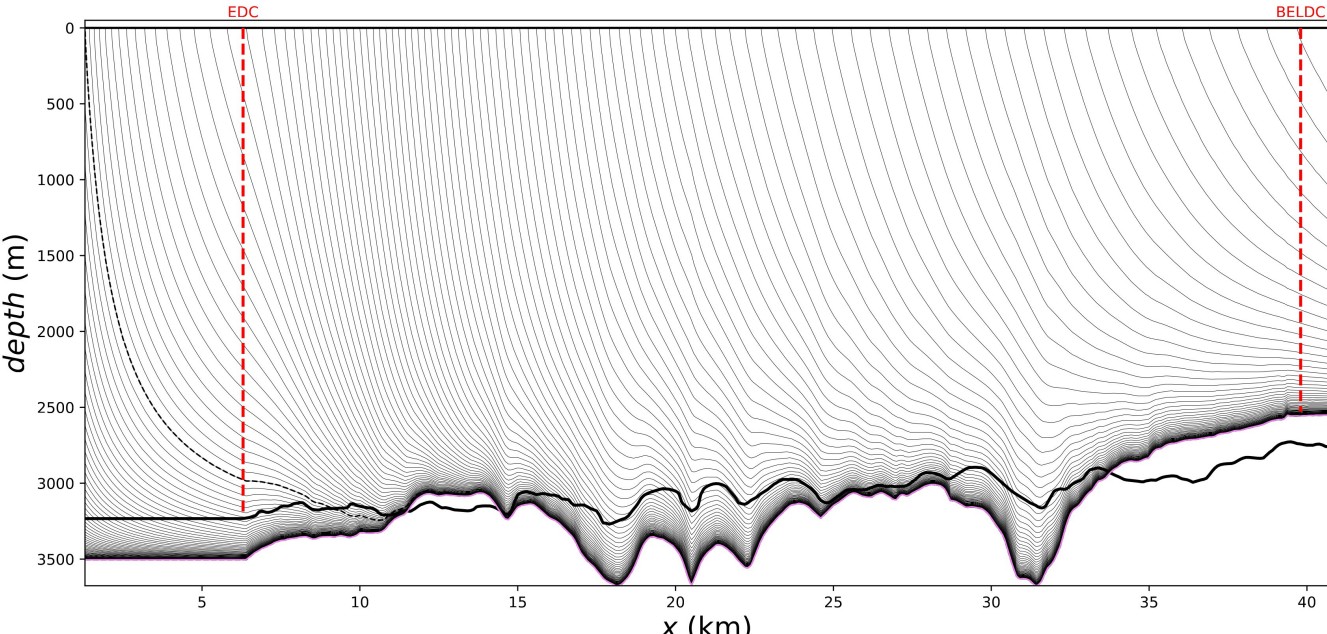

**Figure 8:** Trajectories of ice particles (black lines) along the DC-LDC flow line. The positions of the EDC and BELDC deep drill sites are plotted in dashed red. The black dashed line represents the trajectory originating from the surface upstream corner. The observed bedrock is in thick black and the mechanical bedrock in violet. This figure was automatically generated by the age_flow_line-1.0 software.



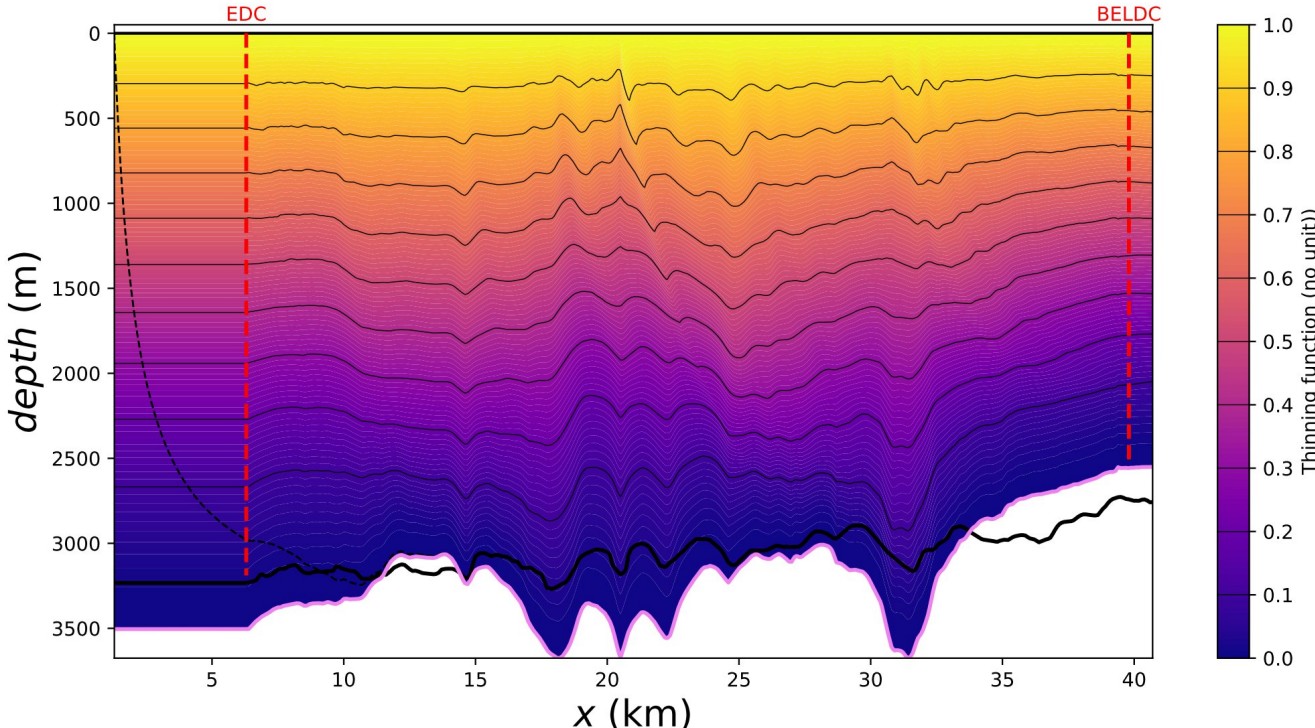

**Figure 9:** Value of the vertical thinning function along the DC-LDC flow line. The two vertical red dashed lines represent the positions of the EDC and BELDC drill sites. The black dashed line represents the trajectory originating from the surface upstream corner. The observed bedrock is in thick black and the mechanical bedrock in violet. This figure was automatically generated by the age_flow_line-1.0 software.




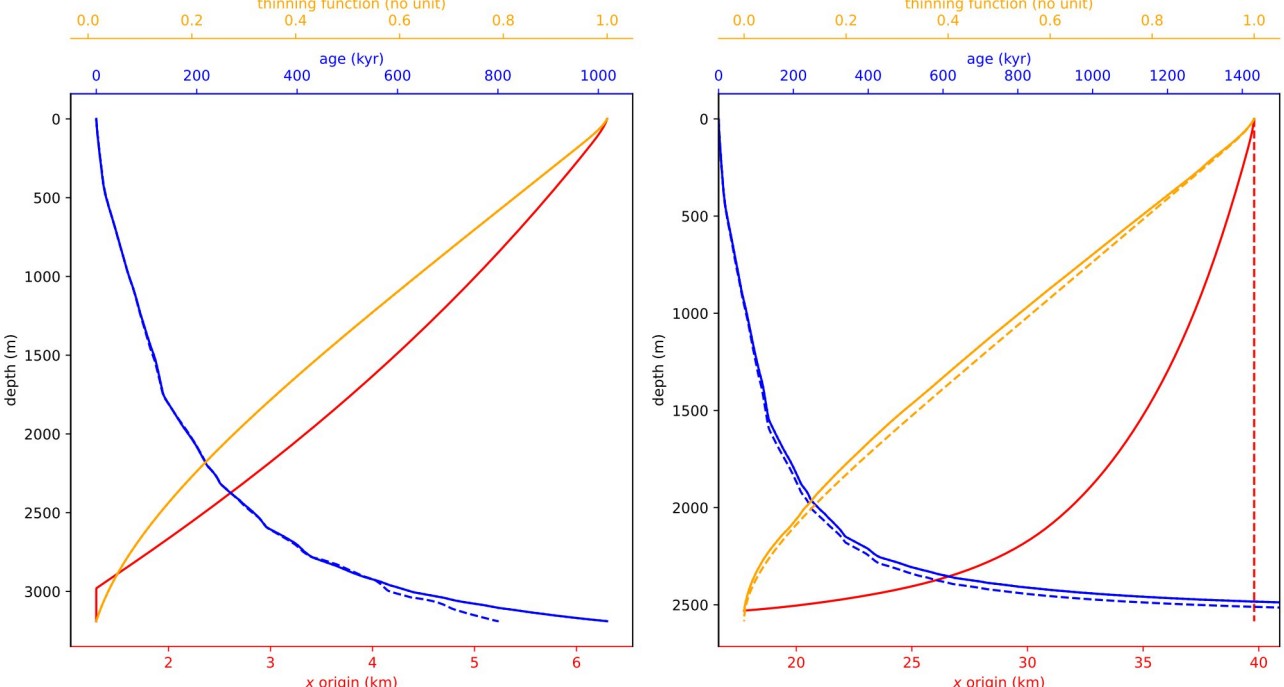

**Figure 10:** Age (blue), vertical thinning function (green) and spatial origin (red) of the ice at the EDC (**left**) and BELDC (**right**) drill site
locations. The solid lines represent the 2.5D age_flow_line-1.0 results. The dashed lines represent the AICC2012 chronology for EDC and
the results of the 1D model (Chung et al., 2023) for BELDC. These figures were automatically generated by the age_flow_line-1.0 software.