# Peer review of "age\_flow\_line-1.0: a fast and accurate numerical age model for a pseudo-steady flow tube of an ice sheet"

_EGUsphere, 2024_

## Referee Comment (RC2)

**Review of "age_flow_line-1.0: a fast and accurate numerical age model for a pseudo-steady flow tube of an ice sheet" by Parrenin et al. 2025**

**General Comments**

This paper presents a 2.5D Eulerian-Lagrangian age model for ice-sheet stratigraphy, assuming steady-state geometry along a flow tube. Using the $(\pi,\phi)$ coordinate system introduced by Parrenin et al. (2006), the model efficiently and accurately solves the transport equations. The manuscript is well within the scope of *Geoscientific Model Development (GMD)*, and the figures are clear and helpful. Overall, I recommend this paper for publication.

That said, the manuscript could more clearly articulate the novelty of this work. The introduction would benefit from a broader contextualization of ice-sheet stratigraphy and dating methods, clarifying when modeling is necessary compared to alternative approaches such as layer counting or tephra dating. When discussing existing models, it would be helpful to distinguish between different objectives, such as dating deep ice-core layers, determining ice origin for upstream corrections, modeling the age distribution across an ice sheet, or using isochrones to invert for basal parameters. A brief discussion of key challenges in ice-age modeling, such as numerical diffusion, would also be valuable. Strengthening the motivation of why a new age model is necessary and how this model fills a gap in the existing literature would further improve the framing of the study.

The application of the model to simulate the age distribution between Dome C and Little Dome C is a helpful illustration, but the purpose of this demonstration could be clearer. The paper initially suggests that the goal is to investigate the role of horizontal advection in shaping the depth-age relationship at Little Dome C, yet the model parametrization is later described as unrealistic. This raises questions about the extent to which conclusions drawn from this 'unrealistic' model run are meaningful. It would be helpful to disentangle these two aspects: Is the aim to assess the significance of horizontal advection, or simply to demonstrate the model's capabilities? Clarifying these objectives, as well as the connection to Chung et al. (2024), would strengthen the narrative.

The discussion is thorough but could better address model validation, limitations, and potential applications. The role of horizontal flow is highlighted, yet discrepancies between modeled and observed isochrones are only briefly mentioned. Expanding on possible sources of error and how the model compares to other 2.5D approaches (e.g., Buchardt et al. 2007) would be beneficial. Additionally, while the authors acknowledge simplifications in accumulation and basal melt assumptions, a more critical discussion of potential biases would strengthen the manuscript.

The code is well-structured and documented, and I was able to run the DC-BELDC example without issues. Expanding the GitHub README with guidance on adapting the code to other regions would be beneficial. While Section 2.4 provides useful information for users, its placement disrupts the readability of the manuscript. Consider moving it to the GitHub repository for better accessibility.

**Specific Comments**

Line 12: Specify the coordinate system more clearly—replace "innovative" with "logarithmic flux coordinate system."

Lines 21-24: The sentence is too long; consider breaking it down or simplifying.

Lines 26-28: This sentence is hard to follow. Consider restructuring for clarity.

Line 29: Add a sentence elaborating on why this is important or provide an example of how upstream effects impact the paleo-record.

Line 30: Clarify that modeling is particularly important where annual layer counting is not possible.

Lines 32-35: This sentence is difficult to understand. Aren't you constraining the *boundary conditions of ice sheet models* and understand *internal processes of the ice sheet*? It would also be good to specify what is being inverted for and give an example with citation

Line 36: "types". Implemented into what?

Line 37: Specify what they are used for.

Line 37: Born & Robinson (2021) could also be relevant here.

Lines 57-62: Explain the motivation for developing a new age model—what improvement does it offer over existing models?

Line 58: Wasn't this already introduced in Parrenin et al. (2006)?

Line 65: Consider specifying "steady-state flow-tube." Indicate that the flow line starts at an ice-sheet dome and ends at the margin (since x is later defined as "distance from the dome").

Line 70: Specify whether z is defined as positive (height above bed/sea level) or negative (depth below surface).

Line 84: "Passing below depth z"—note that depth was previously defined as "d", but I'm not sure that 'd' is actually used. Ensure consistency.

Line 113:  How do you justify that basal melt rate and surface accumulation have the same temporal variation? There is no direct response of basal melt rate to accumulation changes.

Line 115: Change to "temporal average."

Line 122: For a general model description, reword as: "using a relative density profile informed by ice core observations or firn models in the simulated area."

Line 132-133: 'a point slightly downstream of the dome' could be more specific. How far downstream?

Line 138: "linear-by-parts function."

Line 141: Delete "that" after "1/a."

Lines 158-166 & 174-178: Consider removing these paragraphs—it sounds more suited for a "README" in the GitHub repository rather than the paper.

Line 179: Clarify—do you mean "flow tube"?

Lines 180-181: Provide more context about Beyond Epica for readers unfamiliar with the project. For example that modeling here is necessary for ice core dating due to small layer thicknesses. It would also be beneficial to already here explain the motivation for this modeling effort.

Lines 185-186: Clarify why you use "mechanical ice thickness" for the bottom boundary conditions. Do you mean that you are using basal conditions from Chung et al. (2023)? Why not use observed ice thickness from your radar survey?

Lines 186-187: The aim of the simulation should be stated earlier in the paragraph rather than after explaining model parameterization. The purpose is unclear—you state that you investigate whether horizontal advection affects depth-age relationships at LDC but then call the setup unrealistic. Separate these two points and clarify the goal here versus in Chung et al. (2024).

Line 191: Where do these boundary conditions come from?

Line 193: BELDC has not been defined previously.

Line 201, Fig. 8: Indicate where the ice divide is located. This would make the statement "the ice particles may originate >20 km upstream along the divide" clearer.

Line 221: Cite as "1D inverse model by Chung et al. (2023)."

Lines 230-231: Missing citations for this statement —add references.

Line 234: "In front of"—do you mean "compared to"?

Line 235: what do you mean with "right" here? How do you know what's right without direct observations.

Line 254: Sentence is incomplete—revise for clarity.

Figure 6: Explain the cause of discrepancies between observed and modeled bedrock.

Line 345: The citation of this thesis seems out of place. Does most of the content of this paper rely on it? Consider revising.

Figure 1: Clarify whether z points upwards or downwards.

Figure 3: A map overview of where this is located in Antarctica could be useful. Instead of surface elevation, I would consider the surface flow velocities a more relevant context for this work.

Figure 7: Consider a logarithmic colormap

Figure 10: I don't see a green line.

**Technical comments**

Line 66: e.g.
Line 195: In Fig. 5.

Line 320, 322, 328, 335: missing DOIs.

Site and Ice Core Naming Consistency: The distinction between Dome C and Little Dome C as sites, and EDC/BELDC as ice core names, may be confusing for readers unfamiliar with the terminology. Consider using a single consistent name per site throughout the text. Right now it is a

mix of DC, EDC, Dome C and LDC, Little Dome C, BELDC, and Beyond EPICA. Alternatively, explicitly define all terms early in the paper.

---

## Author Comment (AC1)

Parrenin et al. report a 2.5D "flowtube" model that utilizes a coordinate transformation that greatly improves the numerical efficiency. This coordinate transformation was developed in previous publications roughly a decade ago, so the primary new aspect of this work is providing the model code so that others can more easily use the model. This paper is well suited for Geophysical Model Development.

The manuscript is clearly written and the primary equations and assumptions are well described and justified. The figures are informative, and are mostly auto-generated from the code. There is no particular scientific conclusion to the paper, which is ok since that is not the primary purpose. The application to EDC-BELDC is appropriate and demonstrates the model capabilities.

I have used this model before and found it useful, functional, and well documented.

Thank you very much for your careful and constructive reviewing work on our manuscript.

I have only a few suggestions given below:

- The conclusion is missing text and should be expanded upon.

We have expanded the conclusion:

**5 Conclusions**

For the interpretation of radar-observed isochrones or ice core chronologies, it is sometimes necessary to simulate the age of the ice in an ice sheet. We have developed a numerical model to calculate the age of the ice along a pseudo-steady flow tube of an ice sheet. Our Eulerian-Lagrangian scheme combines advantages of Eulerian and Lagragian schemes. There is a regular

12

grid where the age is calculated, as in Eulerian schemes, which is significantly more convenient than having to define initial particle positions and tracking these positions during the time evolution. But at the same time, and there is no numerical diffusion, as in Lagrangian schemes, and our model is numerically very accurate. Our model is computationally With our assumption that 1/a and that z_Ω and linear piecewsie functions effective, which opensopening up new prospects for optimizing its parameters according to observations, which requires many forward simulations runs. We have applied our model to the DC-BELDC flow line and we have shown that horizontal flow is non negligible there, with ice particles sometimes travelling >15 km which has implications for the age scale of the Beyond EPICABELDC ice core. The next step is to optimize the parameters of the model according to age observations such as radar-observed isochrones and ice core dated horizons, which is done in the companion manuscript, Chung et al. (2024).

- In the abstract, intro, and conclusion, the coordinate transformation should be described with an additional sentence. What is the gist of the coordinate transformation?

We now refer to this coordinate system as "logarithmic flux" coordinate system and specify that it tracks ice trajectories.

- L21 - change "most important" to "largest" since "most important" is an opinion

Done.

- L22-25 - give references for each of these points and separate with semicolons rather than commas

Done.

- L26 - make "type" plural

We think you referred to L36 and we corrected to "types".

- L50 - make "scheme" plural

Corrected.

- L234 - change "in front of" to "compared to"

Corrected.

- L256 - change ">15km" to ">20km" to be consistent with other locations in the paper

Changed every occurrence to ">15 km" since it is the most correct estimate.

- Figure 1. I don't understand the labeling of "Q(x)" beneath the ice sheet, should it be m(x)? The caption could also use more description of what the symbols represent.

This originally was "Q_m(x)", the melting flux, but a bug in our software skipped the subscript during pdf export. This is now corrected.

- Figure 5. Can you describe why the red dashed lines in the top panel for the core sites don't reach the bottom of the graph? I think this is because the model domain gets to older ages than is actually found at the ice core sites, but it isn't clear.

We added this sentence to the caption:

**Figure 5:** Mesh of the age_flow_line-1.0 model experiment in the $(\pi,\theta)$ (**top**) and (x,z) (**bottom**) coordinate system. The positions of the EDC and BELDC deep drill sites are plotted in dashed red. The observed bedrock is in thick black and the mechanical bedrock in violet. For better readability, the resolution of the grids has been decreased by a factor of 5. Note that in the top panel, the EDC and BELDC ice cores do not extend down to the bottom of the mesh, since this mesh converges asymptotically towards the mechanical bedrock but never reaches it. These figures were automatically generated by the age_flow_line-1.0 software.

- Figure 6 - I think added subpanels with the horizontal flux shape function plotted for each core site would make the figure more interpretable

We added the Ω flux shape function for the two ice cores in Figure 10.

- Figure 9 - mention in Figure 9 caption that the vertical thinning functions at EDC and BELDC are shown in Figure 10

Done.

---

## Author Comment (AC2)

Review of "age_flow_line-1.0: a fast and accurate numerical age model for a pseudo-steady flow tube of an ice sheet" by Parrenin et al. 2025

General Comments

This paper presents a 2.5D Eulerian-Lagrangian age model for ice-sheet stratigraphy, assuming steady-state geometry along a flow tube. Using the (π,φ) coordinate system introduced by Parrenin et al. (2006), the model efficiently and accurately solves the transport equations. The manuscript is well within the scope of Geoscientific Model Development (GMD), and the figures are clear and helpful. Overall, I recommend this paper for publication.

We thank you very much for your careful, detailed and constructive reviewing work on our manuscript.

That said, the manuscript could more clearly articulate the novelty of this work. The introduction would benefit from a broader contextualization of ice-sheet stratigraphy and dating methods, clarifying when modeling is necessary compared to alternative approaches such as layer counting or tephra dating. When discussing existing models, it would be helpful to distinguish between different objectives, such as dating deep ice-core layers, determining ice origin for upstream corrections, modeling the age distribution across an ice sheet, or using isochrones to invert for basal parameters. A brief discussion of key challenges in ice-age modeling, such as numerical diffusion, would also be valuable. Strengthening the motivation of why a new age model is necessary and how this model fills a gap in the existing literature would further improve the framing of the study.

We modified the following paragraph:

Modelling the age and trajectories of the ice in ice sheets is important for several reasons. This includes dating existing ice cores and correcting from upstream effects (Buchardt, 2009; Huybrechts et al., 2007; Johnsen and Dansgaard, 1992; Parrenin et al., 2007). Ice cores can also be dated by counting annual layers where these layers are thin enough (Svensson et al., 2008) or by identifying dated horizons (Bouchet et al., 2023). But modelling also provides an estimate of the thinning and horizontal displacement of ice layers. For example, moving down in the EDML ice core, there is a decreasing trend in the ice isotopic record that corresponds to a decrease in atmospheric temperature during snow deposition. This decrease is not due to temporal climatic variations, but simply to the fact that deeper down, the ice originates from further upstream, from a higher elevation and therefore colder site (Huybrecht et al., 2007). and Moreover, investigating new potential ice core drill sites (Chung et al., 2023; Obase et al., 2023; Parrenin et al., 2017; Van Liefferinge and Pattyn, 2013) usually requires some modelling since radar-observed isochrones generally do not cover the whole ice column. Age observations from ice cores (Bouchet et al., 2023; Oyabu et al., 2022) or from radio echo sounding (Cavitte et al., 2016; MacGregor et al., 2015) can also help constrain the boundary conditions of ice sheets or understand the internal processes of ice sheet models using inverse methods. For example, dated isochrones can help deduce the surface accumulation rate, the basal melting rate or the horizontal and vertical velocity profiles (Buchardt et al., 2007; Parrenin et al., 2017; Chung et al., 2023). Finally, modelling is necessary to estimate the age distribution across an ice sheet, since it is usually not covered entirely by observations. Key challenges in age modelling include numerical accuracy and efficiency, the match to observations and accounting for boundary conditions.

The application of the model to simulate the age distribution between Dome C and Little Dome C is a helpful illustration, but the purpose of this demonstration could be clearer. The paper initially suggests that the goal is to investigate the role of horizontal advection in shaping the depth-age relationship at Little Dome C, yet the model parametrization is later described as unrealistic. This raises questions about the extent to which conclusions drawn from this 'unrealistic' model run are meaningful. It would be helpful to disentangle these two aspects: Is

the aim to assess the significance of horizontal advection, or simply to demonstrate the model's capabilities? Clarifying these objectives, as well as the connection to Chung et al. (2024), would strengthen the narrative.

To clarify, we show in the current manuscript that horizontal flow is important along the DC-BELDC profile because we do not simulate the same isochrones with a 1D model (Chung et al., 2023) or with the current 2.5D model using the same set of input parameters. While in this manuscript we describe the forward 2.5D model, Chung et al. (2024) describe the inverse method around this 2.5D model where the parameters are optimized so that the model fits the ice core and isochronal age observations.

We have modified section 4.2 as follows:

**4.2 Modelling of the DC-BELDC flow line**

We have performed a simulation where both vertical and horizontal ice flow is taken into account. We show that horizontal flow has a non negligible effect on the age modelling of the BELDC ice core, since our modelled isochrones do not have the same geometry than the ones simulated by the 1D model (Chung et al., 2023) when using the same set of input parameters. Indeed, while the 1D model simulates isochrones very close to the observed ones (Chung et al., 2023), the

isochrones simulated by the 2.5D model significantly deviate from them (Figure 7). In our forward simulation using parameters inverted by the 1D model,  particles originate sometimes >20 km from their current position. Our simulation is provided as a proof of concept for the model we have developed. However, it is not appropriate to use the result of the 1D inverse model by Chung et al. (2023) to determine the parameters of the 2.5D model. Furthermore, we can see in Figure 7, that modelled isochrones (in black) do not agree well with the observed isochrones (in white). For a more realistic simulation, the parameters of the model should be optimized so that modelled age fits the radar and ice core age observations. This is the purpose of the companion paper of Chung et al. (2024) who developed this inverse methodology.

The discussion is thorough but could better address model validation, limitations, and potential applications. The role of horizontal flow is highlighted, yet discrepancies between modeled and observed isochrones are only briefly mentioned. Expanding on possible sources of error and how the model compares to other 2.5D approaches (e.g., Buchardt et al. 2007) would be beneficial. Additionally, while the authors acknowledge simplifications in accumulation and basal melt assumptions, a more critical discussion of potential biases would strengthen the manuscript.

As for model validation, we compare our results with the 1D model of Chung et al. (2023). Limitations is already covered in sections 4.3 and 4.4 but going beyond that would require to compare with other models which would be a significant work which could be the subject of a future manuscript. Potential applications and comparison to other modelling effort is now covered in a new section 4.5 pasted below:

**4.5 Possible applications and comparison with other modelling efforts¶**

This flow tube model could be applied to several nearly steady flow lines of the current Greenland and Antarctic ice sheets, especially those that have ice core information. ¶

The Vostok flow line in East Antarctica was originally modelled by Ritz (1992) but only the age along the Vostok ice core was calculated by a Lagrangian tracer scheme. This work was later extended by Parrenin et al. (2001; 2004) who optimized some parameters of the model to fit the ice core age observations, but still without accounting for the isochronal information. Another flow tube model was developed and applied to the Vostok flow line by Salamatin et al. (2009), accounting for both the ice core and isochronal age observations. It would be valuable to apply the current numerical model to this Vostok flow line and compare the results with these previous modelling efforts.¶

The EDML ice core in East Antarctica is situated on a ridge and horizontal flow also needs to be taken into account. Huybrecht et al. (2007) applied a large scale ice sheet model of Antarctica with a local high resolution and high order model nested inside around EDML. It would be valuable to see how our numerical model, with its very high resolution and numerical accuracy and its fitting to isochronal observations, compares with this previous transient and large scale model.¶

The NEEM and NorthGRIP ice cores in Greenland are also situated on a ridge coming from Summit. Isochrones around GRIP were simulated using a flow tube model by Buchardt and Dahl-Jensen (2007) and fitted to radar observations. The basal melting was hence deduced. Applying our numerical model to this flow line, we could compare it to the previous work and see if its high resolution and accuracy can improve the modelling scenario.¶

Another interesting flow line in Greenland is the one going from Summit to the EastGRIP ice core. The age of the ice along this profile was modelled by Gerber et al. (2021) using a similar model to that of the NorthGRIP flow line, based on a Dansgaard-Johnsen velocity profile (Johnsen and Dansgaard, 1992). Again, our numerical model could be compared to this previous modelling effort.

The code is well-structured and documented, and I was able to run the DC-BELDC example without issues. Expanding the GitHub README with guidance on adapting the code to other regions would be beneficial. While Section 2.4 provides useful information for users, its placement disrupts the readability of the manuscript. Consider moving it to the GitHub repository for better accessibility.

We added the following sentences in the README:

> If you want to set up a new flow tube experiment, we suggest to copy an existing experiment directory such as DC-BELDC. Then you can incrementally modify the `parameters.yml` parameter file and the `.txt` data files.

As for section 2.4, we considered removing it but we reckon that it is OK to have such a technical section on the code itself for a GMD manuscript which is not only about scientific conclusions, but also tool development.

Specific Comments

Line 12: Specify the coordinate system more clearly—replace "innovative" with "logarithmic flux coordinate system."

"logarithmic flux" added.

Lines 21-24: The sentence is too long; consider breaking it down or simplifying.

We broke it down using semi-columns.

Lines 26-28: This sentence is hard to follow. Consider restructuring for clarity.

We broke this sentence in two sentences for clarity.

Line 29: Add a sentence elaborating on why this is important or provide an example of how upstream effects impact the paleo-record.

We modified this paragraph as follows:

Modelling the age and trajectories of the ice in ice sheets is important for several reasons. This includes dating existing ice cores and correcting from upstream effects (Buchardt, 2009; Huybrechts et al., 2007; Johnsen and Dansgaard, 1992; Parrenin et al., 2007). Ice cores can also be dated by counting annual layers where these layers are thin enough (Svensson et al., 2008) or by identifying dated horizons (Bouchet et al., 2023). But modelling also provides an estimate of the thinning and horizontal displacement of ice layers. For example, moving down in the EDML ice core, there is a decreasing trend in the ice isotopic record that corresponds to a decrease in atmospheric temperature during snow deposition. This decrease is not due to temporal climatic variations, but simply to the fact that deeper down, the ice originates from further upstream, from a higher elevation and therefore colder site (Huybrecht et al., 2007). andMoreover, investigating new potential ice core drill sites (Chung et al., 2023; Obase et al., 2023; Parrenin et al., 2017; Van Liefferinge and Pattyn, 2013) usually requires some modelling since radar-observed isochrones generally do not cover the whole ice column. Age observations from ice cores (Bouchet et al., 2023; Oyabu et al., 2022) or from radio echo sounding (Cavitte et al., 2016; MacGregor et al., 2015) can also help constrain the boundary conditions of ice sheets or understand the internal processes of ice sheet models using inverse methods. For example, dated isochrones can help deduce the surface accumulation rate, the basal melting rate or the horizontal and vertical velocity profiles (Buchardt et al., 2007; Parrenin et al., 2017; Chung et al., 2023). Finally, modelling is necessary to estimate the age distribution across an ice sheet, since it is usually not covered entirely by observations. Key challenges in age modelling include numerical accuracy and efficiency, the match to observations and accounting for boundary conditions.¶

Line 30: Clarify that modeling is particularly important where annual layer counting is not possible.

See above the modified paragraph.

Lines 32-35: This sentence is difficult to understand. Aren't you constraining the boundary conditions of ice sheet models and understand internal processes of the ice sheet? It would also be good to specify what is being inverted for and give an example with citation

We added the following sentence:

boundary conditions of ice sheets or understand the internal processes of ice sheet models using inverse methods. For example, dated isochrones can help deduce the surface accumulation rate, the basal melting rate or the horizontal and vertical velocity profiles (Buchardt et al., 2007; Parrenin et al., 2017; Chung et al., 2023). Finally, modelling is necessary to estimate the age distribution across an ice sheet, since it is usually not covered entirely by observations. Key challenges in age modelling include numerical accuracy and efficiency, the match to observations and accounting for boundary conditions.¶

Line 36: "types". Implemented into what?

"types" corrected. "implemented" changed for "developed".

Line 37: Specify what they are used for.

Sentence changed to:

developedimplemented. For example, large scale transient models have been used (Lhomme et al., 2005a; Sutter et al., 2019, Born and Robinson, 2021) to estimate the stratigraphy of the Greenland and Antarctic ice sheets. The advantages of these

Line 37: Born & Robinson (2021) could also be relevant here.

Reference added.

Lines 57-62: Explain the motivation for developing a new age model—what improvement does it offer over existing models?

We added the following sentence:

scheme that uses an analytical derivation of trajectories and a grid which tracks these trajectories. This model offers improved numerical accuracy and efficiency over existing models and is therefore appropriate for inverse methods where many forward simulations are necessary. In Section 2, we first present the analytical and numerical foundations of the model and its

Line 58: Wasn't this already introduced in Parrenin et al. (2006)?

The coordinate system was published in Parrenin et al. (2006) but only in a restrictive case where the velocity profiles are spatially homogeneous. Parrenin and Hindmarsh (2007) then generalized the approach. We added the first reference as well.

Line 65: Consider specifying "steady-state flow-tube." Indicate that the flow line starts at an ice-sheet dome and ends at the margin (since x is later defined as "distance from the dome").

Sentence modified to:

We first consider a steady flow line tube of an ice sheet, which starts at a dome and ends at the ice sheet margin. This means that we assume that factors such as the geometry of athe flow linetube (e.g. ice thickness) and the vertical shape function do not change in time. In this model, we will consider non-steady accumulation rate and melting rate, but this will be discussed

Line 70: Specify whether z is defined as positive (height above bed/sea level) or negative (depth below surface).

We now specify:

we allow the flow tube width and relative density to vary vertically. We write the equations in the $(x,z)$ coordinates where $x$, the horizontal coordinate along the direction of ice flow, is the distance from the dome and $z$ is the vertical coordinate pointing upward. We suppose that the horizontal direction of the flow does not depend on the vertical position and is time-independent,

Line 84: "Passing below depth z"—note that depth was previously defined as "d", but I'm not sure that 'd' is actually used. Ensure consistency.

d is actually not used so we removed the definition.

We now write:

We now define fluxes used to derive the stream function. The partial horizontal flux $q_H(x,z)$ is defined as the horizontal flux passing below depthelevation $z$:

Line 113: How do you justify that basal melt rate and surface accumulation have the same temporal variation? There is no direct response of basal melt rate to accumulation changes.

*Well, we do not justify this since it is just a mathematical simplification but which has no physical justification. This is explained in the discussion, section 4.3:*

> Second, the same temporal multiplicative factor is applied to both surface accumulation and basal melting. There are no physical reason to assume surface accumulation and basal melting have varied in concert, this is just a mathematical convenience. But because basal melting is generally small compared to surface accumulation, this assumption should not be too dramatic if the  real average-in-time basal melting is given as input to the model. Moreover, this temporal factor assumes that the surface accumulation spatial pattern has remained stable in time. This assumption relies on a stable snow precipitation process and a stable snow re-deposition by wind, which might not always be the case (Cavitte et al., 2018) .

*We now refer to the discussion when introducing this assumption.*

Line 115: Change to "temporal average."

*Done.*

Line 122: For a general model description, reword as: "using a relative density profile informed by ice core observations or firn models in the simulated area."

*Done, thank you for the suggestion.*

Line 132-133: 'a point slightly downstream of the dome' could be more specific. How far downstream?

*We changed these sentences as follow:*

> boundary should be known. As the $\pi$ scale is logarithmic, there is a singularity at the dome. Therefore, the horizontal $\pi$ scale starts at a point slightly downstream of the dome, where an age can be prescribed, for example using a dome solution (which assumes purely vertical movement). If one is interested in a flank ice core, the upstream boundary can be chosen such that the upstream condition does not affect the age modelling of the ice core, i.e., the ice in the ice core comes from the surface boundary and not from the upstream boundary.

Line 138: "linear-by-parts function."

*Corrected.*

Line 141: Delete "that" after "1/a."

*Deleted, thanks.*

Lines 158-166 & 174-178: Consider removing these paragraphs—it sounds more suited for a "README" in the GitHub repository rather than the paper.

*We reckon it is OK to have such a section in a GMD manuscript, which is not only about scientific conclusions, but also about tool availability.*

Line 179: Clarify—do you mean "flow tube"?

*We get the age along a flow line but for that we model a flow tube, so in our opinion both formulations are correct.*

Lines 180-181: Provide more context about Beyond Epica for readers unfamiliar with the project. For example that modeling here is necessary for ice core dating due to small layer

thicknesses. It would also be beneficial to already here explain the motivation for this modeling effort.

We modified this section as follow:

To demonstrate the performance and ability of our age model, we apply it to the East Antarctica flow line between DC and the BELDC drill site . Beyond EPICA is a European project that aims to drill a continuous ice core record back to 1.2 million years at least, which makes this flow line particularly interesting. Numerical modelling is necessary to estimate the age of the ice deeper than the deepest visible radar horizons and to estimate the trajectories of ice particles. The DC-BELDC flow tube has already been determined in the companion paper Chung et al. (2024). It is ~40 km

Lines 185-186: Clarify why you use "mechanical ice thickness" for the bottom boundary conditions. Do you mean that you are using basal conditions from Chung et al. (2023)? Why not use observed ice thickness from your radar survey?

Sentence change to:

thickness and uses the comparison with the observed ice thickness to infer basal conditions. For consistency and the sake of comparison, it is therefore the mechanical ice thickness that we use for our bottom boundary condition here, instead of the observed ice thickness. The aim of our simulation here is  to estimate whether horizontal advection is an important

Lines 186-187: The aim of the simulation should be stated earlier in the paragraph rather than after explaining model parameterization. The purpose is unclear—you state that you investigate whether horizontal advection affects depth-age relationships at LDC but then call the setup unrealistic. Separate these two points and clarify the goal here versus in Chung et al. (2024).

We changed the formulation of this paragraph which was indeed a bit awkward:

observed ice thickness. The aim of our simulation here is  to estimate whether horizontal advection is an important mechanism to take into account in the modelling of the age along this flow line, by comparing the result of 1D and 2.5D models with the same set of parameters. This 2.5D simulation is however not fully optimized. This is because we are using the results of the 1D inverse model for a 2.5D flow tube model, instead of optimizing the 2.5D model directly onto the observed isochrones and ice core datasets. The optimization of this 2.5D model is the scope of the companion paper Chung et al. (2024).

Line 191: Where do these boundary conditions come from?

We added the following sentence:

The boundary conditions of the model are plotted in Figure 4. The accumulation and basal melting rates are taken from the 1D inverse model (Chung et al., 2023), while the flow tube width is calculated from the back-tracking of adjacents flow lines from BELDC (Chung et al., 2024). The meshes of the model are plotted in Figure 5. The horizontal flux shape function ω is plotted

Line 193: BELDC has not been defined previously.

This is now defined in the introduction.

Line 201, Fig. 8: Indicate where the ice divide is located. This would make the statement "the ice particles may originate >20 km upstream along the divide" clearer.

We now write in the figure caption:

**Figure 8:** Trajectories of ice particles (black lines) along the DC-BELDC flow line which is situated along a divide. The positions of the EDC and BELDC deep drill sites are plotted in dashed red. The black dashed line represents the trajectory originating from the surface upstream corner. The observed bedrock is in thick black and the mechanical bedrock in violet. This figure was automatically generated by the age_flow_line-1.0 software.

Line 221: Cite as "1D inverse model by Chung et al. (2023)."

Done.

Lines 230-231: Missing citations for this statement —add references.

Sentence changed to:

in the past, which is still unclear. Greenland and West Antarctica have probably encountered more important changes since they are more sensitive to climatic variations in the ocean and in the atmosphere (Quiquet et al., 2013; Wolff et al., 2025).

Line 234: "In front of"—do you mean "compared to"?

Yes, corrected.

Line 235: what do you mean with "right" here? How do you know what's right without direct observations.

Changed "right" to "real".

Line 254: Sentence is incomplete—revise for clarity.

Yes, sorry, copy/paste problem which is now corrected.

Figure 6: Explain the cause of discrepancies between observed and modeled bedrock.

In this legend, we now specify:

**Figure 6:** The $\omega$ horizontal flux shape function along the DC-BELDC flow line. The positions of the EDC and BELDC deep drill sites are plotted in dashed red. The observed bedrock is in thick black and the mechanical bedrock (the elevation where the extrapolated velocity becomes zero) in violet. This figure was automatically generated by the age_flow_line-1.0 software.

Line 345: The citation of this thesis seems out of place. Does most of the content of this paper rely on it? Consider revising.

In this thesis, we generalized the analytical developments done in Parrenin et al. (2006) and Parrenin and Hindmarsh (2007), so we think it is a useful citation.

Figure 1: Clarify whether z points upwards or downwards.

z points upwards, as we now write in section 2.1. This is consistent with the figure.

Figure 3: A map overview of where this is located in Antarctica could be useful. Instead of surface elevation, I would consider the surface flow velocities a more relevant context for this work.

We modified the figure according to your suggestions:

[Figure]

Figure 7: Consider a logarithmic colormap

We considered a logarithmic colormap, but kept the current colormap for two reasons. First, the age at surface is zero and therefore a logarithmic colormap would have a singularity at surface. Second, we find it useful to visualize that the young layers occupy most of the thickness.

Figure 10: I don't see a green line.

Yes, sorry, corrected to orange.

Technical comments

Line 66: e.g.

Corrected.

Line 195: In Fig. 5.

Corrected.

Line 320, 322, 328, 335: missing DOIs.

322 unfortunately does not have a DOI. The 3 others were corrected.

Site and Ice Core Naming Consistency: The distinction between Dome C and Little Dome C as sites, and EDC/BELDC as ice core names, may be confusing for readers unfamiliar with the terminology. Consider using a single consistent name per site throughout the text. Right now it is a mix of DC, EDC, Dome C and LDC, Little Dome C, BELDC, and Beyond EPICA. Alternatively, explicitly define all terms early in the paper.

Thanks for the comment.
We replaced "LDC" and "Little Dome C" with "BELDC" throughout the manuscript.
We replaced "Beyond EPICA ice core" with "BELDC ice core" throughout the manuscript.
"DC" is now defined once and used instead of Dome C throughout the manuscript.

---

## Referee Report (RR1)

**Reviewer Comments – Round 2**

Thank you for thoroughly addressing my previous comments. The manuscript has improved in both clarity and structure, and I appreciate the authors' thoughtful and detailed responses to the concerns raised in the first round of review.

My only remaining main suggestion concerns **Section 4.5**, where the discussion could be further strengthened by more explicitly situating the model within the context of the listed existing approaches. While I understand that a direct comparison with other models is beyond the scope of this paper, it would be helpful if you could **elaborate on how you think the new method complements or improves upon previous efforts, and what specific advantages or new insights it may offer.**

Aside from this point, I have only a few minor comments, listed below. Once these are addressed, I would be happy to recommend the paper for publication.

Line 34: I think you mean **thick** enough?
Line 43: again, this sentence doesn't make much sense to me. I think what you mean is boundary conditions of **ice sheet models** or understand the internal processes of **ice sheets**  using inverse methods ?

Line 61: a steady-**state** model
Line 95: see my previous comment. Depth is still defined as 'd' but never used. Change to elevation 'z'.
Line 131: How do you actually determine R(t)? please elaborate.
Line 132: Suggested edit for better clarity and text flow:  → The implications of using the same temporal variation for basal melt rate and accumulation rate are discussed in Section 4.3.

Equation (15): replace full stop with comma
Equation (16): missing full stop after equation

Section 3: This is much improved and a lot clearer – Thank you.
Line 223: approach**e**s
Line 248: it would be nice to get some numbers here – e.g. how many meters deviation or depth percentage.
Line 251-252: remove this sentence (repetition of above). There are also no white isochrones in Fig. 7
Line 264-265: "But because basal melting is generally small compared to surface accumulation, this assumption should not be too dramatic" - I'd be careful with this sentence. Basal melt may be small but has quite a large impact on isochrone deformation. I suggest rephrasing to something like:

However, in regions where basal melt rates are low, this assumption should not introduce major errors, as long as a realistic time-averaged basal melt rate is prescribed.

Line 265: I think 'a realistic' would be better suited instead of real

Section 4.5: Thank you for adding this section. It's helpful to see a discussion of past modeling efforts. One element which I feel is still missing, however, is a clearer explanation of how your model might compare to, or improve upon, these previous studies.

While it's fine to outline possible future applications, I think this section would be much more useful if you could briefly articulate where your model offers particular advantages.

From my understanding, one of the strengths of your model lies in producing a high-resolution depth–age distribution and efficient computation, which could be especially valuable for certain applications (long flow lines, old age, inversion..?). However, I am less clear on how well-suited it is for estimating basal conditions in regions with high basal melt, especially since basal melt rate is coupled to accumulation rate in your formulation.

For instance, models by Buchardt & Dahl-Jensen (2007) or Gerber et al. (2021) assume a steady-state basal melt rate and use time-varying accumulation driven by $\delta^{18}O$ climate curves, with some tuning, to estimate basal melt rates. How would you envision your model performing for basal melt estimates in comparison, especially regarding regions where basal melt is considered to be high?

Whatever your interpretation, a few sentences offering your outlook on how this model can advance research — and where its strengths (and potential limitations) lie compared to the available literature — would be a valuable addition to the discussion.

---

## Referee Report (RR2)

[revised manuscript text omitted]

Here we present a new 2.5D (flow tube) pseudo-steady numerical age model, named age\_flow\_line-1.0. This numerical age model uses an innovative coordinate system  $(\pi,\theta)$  previously published (Parrenin et al., 2006; Parrenin and Hindmarsh, 2007) and which is suitable for solving transport equations. We solve the age and transport equations using a Eulerian-Lagrangian scheme that uses an analytical derivation of trajectories and a grid which tracks these trajectories. This model offers improved numerical accuracy and efficiency over existing models and is therefore appropriate for inverse methods where many forward simulations are necessary. In Section 2, we first present the analytical and numerical foundations of the model and its implementation. In Section 3, we show an application of the model to the flow line between Dome C (DC) and the Beyond EPICA Little Dome C (BELDC) ice coring site in Antarctica. In Section 4, we discuss the advantages and limitations of the Does this mean that you choose a control volume and (for positive diversance) to Shrink the tube site accordingly? model.

**2 Description of the model**

80

**2.1 Notations and analytical aspects**

We first consider a steady flow tube of an ice sheet, which starts at a dome and ends at the ice sheet margin. 
[revised manuscript text omitted]
 a relative density profile informed by ice core observations or firn models in the simulated area. Moreover, we assume that the flow tube has vertical walls, that is, Y does not depend on z. This way, the equations for  $\kappa$  and  $\tau$  become:

$$\kappa = \frac{1}{a} \frac{\partial z_{\Omega}}{\partial \Omega}.$$
 (15)

$$\tau = \Omega \frac{a}{a_0} \left( 1 - \frac{1}{\kappa} \int_{\pi_0}^{\pi} \frac{\partial \kappa}{\partial \pi} \right)^{-1}$$

 $\tau = \Omega \frac{a}{a_0} \left( 1 - \frac{1}{\kappa} \int_{\pi_0}^{\pi} \frac{\partial \kappa}{\partial \pi} \right)^{-1}$   $\Delta \text{ so there is adopt at salving for solving any transport equation and an particular adopt at salving the solving any transport equation and an particular adopt at salving the solving and the salving adopt at salving the solving any transport equation and an articular adopt at salving the salving the salving transport equation and an articular adopt at salving the salving transport equation and an articular adopt at salving the salving transport equation and an articular adopt at salving transport equation are also at the salving transport equation and an articular adopt at salving transport equation are also at a salving transport equation and a salving transport equation are also at a salving transport equation and a salving transport equation at a salving transport equation at a salving transport eq$

**2.3 Numerical aspects**

[revised manuscript text omitted]

**3 Application to the DC-BELDC flow line**

200 To demonstrate the performance and ability of our age model, we apply it to the East Antarctica flow line between DC and the BELDC drill site. Beyond EPICA is a European project that aims to drill a continuous ice core record back to 1.2 million years at least, which makes this flow line particularly interesting. Numerical modelling is necessary to estimate the age of the ice deeper than the deepest visible radar horizons and to estimate the trajectories of ice particles. The DC-BELDC flow tube has already been determined in the companion paper Chung et al. (2024). It is ~40 km long with a strong lateral flow divergence because we are on a divide (Figure 3). The parameters used for our simulations, namely the surface accumulation rate, basal 205 melting and Lliboutry parameter, are those from Chung et al. (2023) obtained by fitting a 1D pseudo-steady age model onto observed isochrones. This 1D model finds a best fit value for a mechanical ice thickness and uses the comparison with the observed ice thickness to infer basal conditions. For consistency and the sake of comparison, it is therefore the mechanical ice thickness that we use for our bottom boundary condition here, instead of the observed ice thickness. The aim of our simulation 210 here is to estimate whether horizontal advection is an important mechanism to take into account in the modelling of the age along this flow line, by comparing the result of 1D and 2.5D models with the same set of parameters. This 2.5D simulation is however not fully optimized. This is because we are using the results of the 1D inverse model for a 2.5D flow tube model, instead of optimizing the 2.5D model directly onto the observed isochrones and ice core datasets. The optimization of this 2.5D model is the scope of the companion paper Chung et al. (2024).

The boundary conditions of the model are plotted in Figure 4. The accumulation and basal melting rates are taken from the 1D inverse model (Chung et al., 2023), while the flow tube width is calculated from the back-tracking of adjacents flow lines from BELDC (Chung et al., 2024). The meshes of the model are plotted in Figure 5. The horizontal flux shape function ω is plotted in Figure 6. The modelled age, trajectories and vertical thinning function are plotted in Figure 7, Figure 8 and Figure 9. The

results for the next ice core at BELDC are plotted in Figure 10. These figures (from 4 to 10) are automatically generated by the age\_flow\_line-1.0 software.

Figure 5 shows that the mesh is refined near the bedrock. This is expected, since the grid is exponential with respect to the flux shape function  $\Omega$ . When there is no basal melting (which is the case in our application with the mechanical ice thickness), the grid does not extend down to the bedrock but approachs it asymptotically. The horizontal resolution also varies in our application, but several parameters affect this resolution. One would expect the horizontal resolution to decrease when the distance to the dome increases, since the grid is exponential in total flux Q. But this is partially compensated by the exponential increase of the tube width along the flow line.

Figure 8 shows that the ice particles may originate >15 km upstream along the divide. Therefore, horizontal flow is not negligible along this flow line and should be taken into account when modelling the age of the ice. In particular at BELDC (Figure 10), the bottom ice originates up to ~22 km upstream from our modelling.

**230 4 Discussion**

220

225

240

245

**4.1 Numerical aspects**

This age model is accurate and with minimal numerical diffusion since quantities are transported from one node to the next without interpolation. This could help test the accuracy of numerical schemes of more complex 3D and transient models.

This model has similar complexity of a steady model but considers the variations of accumulation with time which is non-negligible along a glacial-interglacial cycle (approximately a factor of 2 on the East Antarctic plateau). For a  $1000 \times 1000$  grid, the computation time is  $\sim 0.1$  s on a recent laptop computer for the numerical calculations, with a few additional seconds for the building of the figures. For this grid, the memory footprint of the simulation is  $\sim 0.8$  Gb.

Due to its fast computation time, this model is appropriate for inverse simulations where the results are fitted to observations. Indeed, inverse simulations require multiple runs of the forward model to find the optimal set of parameters, which can be computationally expensive.

This model could also be appropriate for representing the advection term in a heat equation, although the diffusion term is better dealt with in a physical (x,z) coordinate system.

**4.2 Modelling of the DC-BELDC flow line**

We have performed a simulation where both vertical and horizontal ice flow is taken into account. We show that horizontal flow has a non negligible effect on the age modelling of the BELDC ice core, since our modelled isochrones do not have the same geometry than the ones simulated by the 1D model (Chung et al., 2023) when using the same set of input parameters. Indeed, while the 1D model simulates isochrones very close to the observed ones (Chung et al., 2023), the isochrones simulated

by the 2.5D model significantly deviate from them (Figure 7). In our forward simulation using parameters inverted by the 1D model, particles originate sometimes >15 km from their current position. Our simulation is provided as a proof of concept for the model we have developed. However, it is not appropriate to use the result of the 1D inverse model by Chung et al. (2023) to determine the parameters of the 2.5D model. Furthermore, we can see in Figure 7, that modelled isochrones (in black) do not agree well with the observed isochrones (in white). For a more realistic simulation, the parameters of the model should be optimized so that modelled age fits the radar and ice core age observations. This is the purpose of the companion paper of Chung et al. (2024) who developed this inverse methodology.

**255 4.3 Limitations of the pseudo-steady assumption**

250

260

265

275

Our model relies on the pseudo-steady assumption, which states that all variables are in steady-state except for a temporal multiplicative factor which is both applied to accumulation and melting.

First, the geometry is assumed to be in steady-state. This assumption is adapted for the interior of East Antarctica, where relative variations in ice thickness were small (Ritz et al., 2001), but it still requires that the flow lines have not varied too much in the past, which is still unclear. Greenland and West Antarctica have probably encountered more important changes since they are more sensitive to climatic variations in the ocean and in the atmosphere (Quiquet et al., 2013; Wolff et al., 2025).

Second, the same temporal multiplicative factor is applied to both surface accumulation and basal melting. There are no physical reason to assume surface accumulation and basal melting have varied in concert; this is just a mathematical convenience. But because basal melting is generally small compared to surface accumulation, this assumption should not be too dramatic if the real average-in-time basal melting is given as input to the model. Moreover, this temporal factor assumes that the surface accumulation spatial pattern has remained stable in time. This assumption relies on a stable snow precipitation process and a stable snow re-deposition by wind, which might not always be the case (Cavitte et al., 2018).

Our pseudo-steady model can therefore be seen as a first approach to model the age field along flow lines at high resolution, but more complex 3D and transient models are necessary to fully investigate the age of the ice in ice sheets.

**270 **4.4 Possible improvements**

In this implementation, we have made a few simplifying assumptions that could be relaxed in the future. First we assumed that the flow tube has vertical walls, which is for example not the case along a ridge (Passalacqua et al., 2016). Second, we converted real depths into ice equivalent depth using a unique density profile along the flow line, while clearly for long flow lines the density profile might change with the distance from the dome. Third, we used a Lliboutry profile for the horizontal flux shape function  $\omega$ , while other profiles could be used. In particular at a steady dome where Raymond arches develop (Raymond, 1983), the Lliboutry profile does not seem to be suitable (Martín and Gudmundsson, 2012). Fourth, we prescribed a dome solution on the upstream boundary of the domain, but any boundary condition could be prescribed.

Due to the numerical simplicity of the model, it may be possible to derive its Jacobian, which would make inverse simulations more efficient.

**280 4.5 Possible applications and comparison with other modelling efforts**

This flow tube model could be applied to several nearly steady flow lines of the current Greenland and Antarctic ice sheets, especially those that have ice core information.

The Vostok flow line in East Antarctica was originally modelled by Ritz (1992) but only the age along the Vostok ice core was calculated by a Lagrangian tracer scheme. This work was later extended by Parrenin et al. (2001; 2004) who optimized some parameters of the model to fit the ice core age observations, but still without accounting for the isochronal information. Another flow tube model was developed and applied to the Vostok flow line by Salamatin et al. (2009), accounting for both the ice core and isochronal age observations. It would be valuable to apply the current numerical model to this Vostok flow line and compare the results with these previous modelling efforts.

The EDML ice core in East Antarctica is situated on a ridge and horizontal flow also needs to be taken into account. Huybrecht et al. (2007) applied a large scale ice sheet model of Antarctica with a local high resolution and high order model nested inside around EDML. It would be valuable to see how our numerical model, with its very high resolution and numerical accuracy and its fitting to isochronal observations, compares with this previous transient and large scale model.

The NEEM and NorthGRIP ice cores in Greenland are also situated on a ridge coming from Summit. Isochrones around NorthGRIP were simulated using a flow tube model by Buchardt and Dahl-Jensen (2007) and fitted to radar observations. The basal melting was hence deduced. Applying our numerical model to this flow line, we could compare it to the previous work and see if its high resolution and accuracy can improve the modelling scenario.

Another interesting flow line in Greenland is the one going from Summit to the EastGRIP ice core. The age of the ice along this profile was modelled by Gerber et al. (2021) using a similar model to that of the NorthGRIP flow line, based on a Dansgaard-Johnsen velocity profile (Johnsen and Dansgaard, 1992). Again, our numerical model could be compared to this previous modelling effort.

**5 Conclusions**

285

290

295

300

305

For the interpretation of radar-observed isochrones or ice core chronologies, it is sometimes necessary to simulate the age of the ice in an ice sheet. We have developed a numerical model to calculate the age of the ice along a pseudo-steady flow tube of an ice sheet. Our Eulerian-Lagrangian scheme combines advantages of Eulerian and Lagragian schemes. There is a regular grid where the age is calculated, as in Eulerian schemes, which is significantly more convenient than having to define initial particle positions and tracking these positions during the time evolution. But at the same time, there is no numerical diffusion,

as in Lagrangian schemes, and our model is numerically very accurate. Our model is computationally effective, opening up new prospects for optimizing its parameters according to observations, which requires many forward simulations runs. We have applied our model to the DC-BELDC flow line and we have shown that horizontal flow is non negligible there, with ice particles sometimes travelling >15 km which has implications for the age scale of the BELDC ice core. The next step is to optimize the parameters of the model according to age observations such as radar-observed isochrones and ice core dated horizons, which is done in the companion manuscript, Chung et al. (2024).

**Code availability**

[revised manuscript text omitted]

- Parrenin, F., Rémy, F., Ritz, C., Siegert, M. J., and Jouzel, J.: New modeling of the Vostok ice flow line and implication for the glaciological chronology of the Vostok ice core, Journal of Geophysical Research: Atmospheres, 109, https://doi.org/10.1029/2004JD004561, 2004.
  - Parrenin, F., Cavitte, M. G. P., Blankenship, D. D., Chappellaz, J., Fischer, H., Gagliardini, O., Masson-Delmotte, V., Passalacqua, O., Ritz, C., Roberts, J., Siegert, M. J., and Young, D. A.: Is there 1.5-million-year-old ice near Dome C, Antarctica?, The Cryosphere, 11, 2427–2437, https://doi.org/10.5194/tc-11-2427-2017, 2017.
- Passalacqua, O., Gagliardini, O., Parrenin, F., Todd, J., Gillet-Chaulet, F., and Ritz, C.: Performance and applicability of a 2.5-D ice-flow model in the vicinity of a dome, Geoscientific Model Development, 9, 2301–2313, https://doi.org/10.5194/gmd-9-2301-2016, 2016.
  - Quiquet, A., Ritz, C., Punge, H. J., and Salas y Mélia, D.: Greenland ice sheet contribution to sea level rise during the last interglacial period: a modelling study driven and constrained by ice core data, Climate of the Past, 9, 353–366,
- 440 https://doi.org/10.5194/cp-9-353-2013, 2013.

- Raymond, C. F.: Deformation in the vicinity of ice divides, J. Glaciol., 29, 357–373, 1983.
- Reeh, N. and Paterson, W. S. B.: Application of a Flow Model to the Ice-divide Region of Devon Island Ice Cap, Canada, Journal of Glaciology, 34, 55–63, https://doi.org/10.3189/S0022143000009060, 1988.Ritz, C.: Un modèle thermomécanique d'évolution pour le bassin glaciaire antarctique Vostok-glacier Byrd: sensibilité aux valeurs des paramètres mal connus, Thèse d'état, Univ. J. Fourier, Grenoble, France, 1992.
- Ritz, C., Rommelaere, V., and Dumas, C.: Modeling the evolution of Antarctic ice sheet over the last 420,000 years: Implications for altitude changes in the Vostok region, J. Geophys. Res., 106, 31943–31964, https://doi.org/10.1029/2001JD900232, 2001.
- Rybak, O. and Huybrechts, P.: A comparison of Eulerian and Lagrangian methods for dating in numerical ice-sheet models, Annals of Glaciology, 37, 150–158, https://doi.org/10.3189/172756403781815393, 2003.
  - Salamatin, A. N., Tsyganova, E. A., Popov, S. V., and Lipenkov, V. Ya.: Ice flow line modeling in ice core data interpretation: Vostok Station (East Antarctica), in: Physics of Ice Core Records 2, edited by: Hondoh, T., Hokkaido University Press, Sapporo, 2009.
- Sutter, J., Fischer, H., Grosfeld, K., Karlsson, N. B., Kleiner, T., Van Liefferinge, B., and Eisen, O.: Modelling the

  Antarctic Ice Sheet across the mid-Pleistocene transition implications for Oldest Ice, The Cryosphere, 13, 2023–2041,

  https://doi.org/10.5194/tc-13-2023-2019, 2019.
  - Sutter, J., Fischer, H., and Eisen, O.: Investigating the internal structure of the Antarctic ice sheet: the utility of isochrones for spatiotemporal ice-sheet model calibration, The Cryosphere, 15, 3839–3860, https://doi.org/10.5194/tc-15-3839-2021, 2021.

- Van Liefferinge, B. and Pattyn, F.: Using ice-flow models to evaluate potential sites of million year-old ice in Antarctica, Clim. Past, 9, 2335–2345, https://doi.org/10.5194/cp-9-2335-2013, 2013.
  - Vaughan, D.G., J.C. Comiso, I. Allison, J. Carrasco, G. Kaser, R. Kwok, P. Mote, T. Murray, F. Paul, J. Ren, E. Rignot, O. Solomina, K. Steffen and T. Zhang, 2013: Observations: Cryosphere. In: Climate Change 2013: The Physical Science Basis. Contribution of Working Group I to the Fifth Assessment Report of the Intergovernmental Panel on Climate Change
- [Stocker, T.F., D. Qin, G.-K. Plattner, M. Tignor, S.K. Allen, J. Boschung, A. Nauels, Y. Xia, V. Bex and P.M. Midgley (eds.)]. Cambridge University Press, Cambridge, United Kingdom and New York, NY, USA.
  Waddington, E. D., Neumann, T. A., Koutnik, M. R., Marshall, H.-P., and Morse, D. L.: Inference of accumulation-rate patterns from deep layers in glaciers and ice sheets, Journal of Glaciology, 53, 694–712, https://doi.org/10.3189/002214307784409351, 2007.
- Wolff, E. W., Mulvaney, R., Grieman, M. M., Hoffmann, H. M., Humby, J., Nehrbass-Ahles, C., Rhodes, R. H., Rowell, I. F., Sime, L. C., Fischer, H., Stocker, T. F., Landais, A., Parrenin, F., Steig, E. J., Dütsch, M., and Golledge, N. R.: The Ronne Ice Shelf survived the last interglacial, Nature, 1–5, https://doi.org/10.1038/s41586-024-08394-w, 2025.

**Figure 2**: Diagram representing the  $(\pi,\theta)$  grid (in black) and the trajectories (blue arrows) in this coordinate system.

**Figure 3:** Map of the flow line (central blue line) going from DC to BELDC with the width of the flow tube according to the blue colour bar. The background colour represents the surface elevation. The grey and pink arrows represent surface velocity measurements. Adapted from Chung et al. (2024).

**Figure 4:** Boundary conditions of the model along the DC-BELDC flow line: surface accumulation rate (green), basal melt rate (blue) and tube width (black). Note that basal melting rate is zero, as a consequence of using a mechanical ice thickness. The position of the EDC and BELDC drilling sites are indicated in red on the top bar. This figure was automatically generated by the age\_flow\_line-1.0 software.

**Figure 5:** Mesh of the age\_flow\_line-1.0 model experiment in the  $(\pi,\theta)$  (**top**) and (x,z) (**bottom**) coordinate system. The positions of the EDC and BELDC deep drill sites are plotted in dashed red. The observed bedrock is in thick black and the mechanical bedrock in violet. For better readability, the resolution of the grids has been decreased by a factor of 5. Note that in the top panel, the EDC and BELDC ice cores do not extend down to the bottom of the mesh, since this mesh converges asymptotically towards the mechanical bedrock but never reaches it. These figures were automatically generated by the age\_flow\_line-1.0 software.

Figure 6: The  $\omega$  horizontal flux shape function along the DC-BELDC flow line. The positions of the EDC and BELDC deep drill sites are plotted in dashed red. The observed bedrock is in thick black and the mechanical bedrock (the elevation where the extrapolated velocity becomes zero) in violet. This figure was automatically generated by the age\_flow\_line-1.0 software.

Figure 7: Modelled age along the DC-BELDC flow line, according to the colour scale on the right. The modelled isochrones are plotted in solid black and their age is represented on the colour bar. The positions of the EDC and BELDC deep drill sites are plotted in dashed red. The black dashed line represents the trajectory originating from the surface upstream corner. The dotted black lines are the isochrones observed by radar. The observed bedrock is in thick black and the mechanical bedrock in violet. This figure was automatically generated by the age\_flow\_line-1.0 software.

**Figure 8:** Trajectories of ice particles (black lines) along the DC-BELDC flow line which is situated along a divide. The positions of the EDC and BELDC deep drill sites are plotted in dashed red. The black dashed line represents the trajectory originating from the surface upstream corner. The observed bedrock is in thick black and the mechanical bedrock in violet. This figure was automatically generated by the age\_flow\_line-1.0 software.

**Figure 9:** Value of the vertical thinning function along the DC-BELDC flow line. The two vertical red dashed lines represent the positions of the EDC and BELDC drill sites. The black dashed line represents the trajectory originating from the surface upstream corner. The observed bedrock is in thick black and the mechanical bedrock in violet. The thinning functions for EDC and BELDC are shown in Figure 10. This figure was automatically generated by the age flow line-1.0 software.

**Figure 10:** Age (blue), vertical thinning function (orange),  $\Omega$  flux shape function (violet) and spatial origin (red) of the ice at the EDC (**left**) and BELDC (**right**) drill site locations. The solid lines represent the 2.5D age\_flow\_line-1.0 results. The dashed lines represent the AICC2012 chronology for EDC and the results of the 1D model (Chung et al., 2023) for BELDC. Note that for EDC, the orange and violet solid lines are superimposed. These figures were automatically generated by the age\_flow\_line-1.0 software.

---

## Author Response (AR2)

**Reviewer 1**

Reviewer Comments – Round 2

Thank you for thoroughly addressing my previous comments. The manuscript has improved in both clarity and structure, and I appreciate the authors' thoughtful and detailed responses to the concerns raised in the first round of review.

Thank you again for your rigorous and insightful review of our manuscript! This is much appreciated.

My only remaining main suggestion concerns Section 4.5, where the discussion could be further strengthened by more explicitly situating the model within the context of the listed existing approaches. While I understand that a direct comparison with other models is beyond the scope of this paper, it would be helpful if you could elaborate on how you think the new method complements or improves upon previous efforts, and what specific advantages or new insights it may offer.

Thank you for your comment. Yes, this discussion was missing. We address this point below.

Aside from this point, I have only a few minor comments, listed below. Once these are addressed, I would be happy to recommend the paper for publication.

Line 34: I think you mean thick enough?

Corrected, thank you.

Line 43: again, this sentence doesn't make much sense to me. I think what you mean is boundary conditions of ice sheet models or understand the internal processes of ice sheets models using inverse methods?

Corrected, thank you.

Line 61: a steady-state model

Changed 'steady' to 'steady-state' throughout the manuscript.

Line 95: see my previous comment. Depth is still defined as 'd' but never used. Change to elevation 'z'.

Definition of 'd' removed.

Line 131: How do you actually determine R(t)? please elaborate.

We added the following sentence:

R(t) is a positive temporal multiplicative factor which can be determined from ice core data, e.g. using the relationship between the isotopic composition of ice and the surface accumulation rate.

Line 132: Suggested edit for better clarity and text flow: This assumption is discussed in section 4.3. → The implications of using the same temporal variation for basal melt rate and accumulation rate are discussed in Section 4.3.

Suggestion adopted, thank you.

Equation (15): replace full stop with comma

Done, thank you.

Equation (16): missing full stop after equation

Done, thank you.

Section 3: This is much improved and a lot clearer – Thank you.

Thank you!

Line 223: approaches

Corrected, thank you.

Line 248: it would be nice to get some numbers here – e.g. how many meters deviation or depth percentage.

We added:

...the isochrones simulated by the 2.5D model significantly deviate from them (Figure 7), with deviations of >100 m at some places.

Line 251-252: remove this sentence (repetition of above). There are also no white isochrones in Fig. 7

Sentence removed.

Line 264-265: "But because basal melting is generally small compared to surface accumulation, this assumption should not be too dramatic" - I'd be careful with this sentence. Basal melt may be small but has quite a large impact on isochrone deformation. I suggest rephrasing to something like:

However, in regions where basal melt rates are low, this assumption should not introduce major errors, as long as a realistic time-averaged basal melt rate is prescribed.

Thanks for the suggestion, we adopted it.

Line 265: I think 'a realistic' would be better suited instead of real

This is now changed according to your suggestion above.

Section 4.5: Thank you for adding this section. It's helpful to see a discussion of past modeling efforts. One element which I feel is still missing, however, is a clearer explanation of how your model might compare to, or improve upon, these previous studies. While it's fine to outline possible future applications, I think this section would be much more useful if you could briefly articulate where your model offers particular advantages.

From my understanding, one of the strengths of your model lies in producing a high-resolution depth—age distribution and efficient computation, which could be especially valuable for certain applications (long flow lines, old age, inversion..?). However, I am less clear on how well-suited it is for estimating basal conditions in regions with high basal melt, especially since basal melt rate is coupled to accumulation rate in your formulation.

For instance, models by Buchardt & Dahl-Jensen (2007) or Gerber et al. (2021) assume a steady-state basal melt rate and use time-varying accumulation driven by  $\delta^{18}$ O climate curves, with some tuning, to estimate basal melt rates. How would you envision your model performing for basal melt estimates in comparison, especially regarding regions where basal melt is considered to be high?

Whatever your interpretation, a few sentences offering your outlook on how this model can advance research — and where its strengths (and potential limitations) lie compared to the available literature — would be a valuable addition to the discussion.

This is what we added in section 4.5:

The value of this model lies in its high numerical accuracy and very fast computation time. It is appropriate for long flow lines, old age and for parameter inferences. The companion paper of Chung et al. (2024) outlines its interest, since we can optimize its parameters in a couple of minutes on a personal computer. However, this model has physical assumptions which might introduce errors. To overcome this limitation, our model could be used in conjunction with a transient model like the one developed by Buchardt & Dahl-Jensen (2007) or Gerber et al. (2021). For example, our model could provide an initial condition which could be refined using the transient model. It could also provide a 'first guess' and a fast approximation of the Jacobian of the transient model, to speed up the optimization of its parameters.

**Reviewer 2**

We warmly thank the reviewer for his careful reading of our manuscript. We took the comments into account and reckon they helped improve the readability of our manuscript.

Answer to specific comments:

- L. 30: I cannot read the last word on this sentence.
- L. 61: replaced 'steady' with 'steady-state'
- L. 63: 'a class to which the age equation belongs'
- L. 62: suggestion adopted.
- L. 72: The age and spatial origin equations belongs to the transport equations. We replaced 'transport' by 'spatial origin' to be more specific.
- L. 83: removed 'will' as suggested, thank you.
- L. 88: Yes, the definition of the flow tube width is relative.
- L. 93: Added 'local' as suggested, thank you.

- L. 95: Definition of 'd' has now been removed after Rev. 1 remark.
- L. 95: Relative density means it is expressed in % of volume, not mass per volume. We added this precision.
- L. 102: Changed equation to:  $Y^B(x') = Y(x',B(x'))$
- L. 110: Sentence modified.
- L. 114: Parentheses removed as suggested, thank you.
- L. 116: No, accumulation cannot be zero.
- L. 121: Replaced by 'equal in magnitude by opposite in sign'.
- L. 140: It is the section of the tube which is rectangular.
- L. 144: Replaced by: 'is very useful to solve any transport equation, a class to which the age equation belongs'
- L. 178: Suggestion adopted, thank you.
- L. 204: Suggestion adopted, thank you.
- L. 205: Suggestion adopted, thank you.
- L. 221: Changed to 'ice-bedrock interface'.
- L. 229: 'originates 22 km upstream from the drilling site".
- L. 232: Suggestion adopted, thank you.
- L. 244: Correction applied, thank you.
- L. 263: Comma replaced by semicolon, thank you.
- Fig. 1, legend: Correction applied, thank you.